# Maternal dysglycaemia, changes in the infant's epigenome modified with a diet and physical activity intervention in pregnancy: Secondary analysis of a randomised control trial

Elie Antoun[1‡], Negusse T. Kitaba[2‡], Philip Titcombe[3], Kathryn V. Dalrymple[4], Emma S. Garratt[2,5], Sheila J. Barton[3], Robert Murray[2], Paul T. Seed[4], Joanna D. Holbrook[2], Michael S. Kobor[6], David TS Lin[6], Julia L. MacIsaac[6], Graham C. Burdge[2], Sara L. White[4], Lucilla Poston[4‡], Keith M. Godfrey[2,5‡], Karen A. Lillycrop[1,5‡]*, UPBEAT Consortium

1 Biological Sciences, Institute of Developmental Sciences, Faculty of Environmental and Life Sciences, University of Southampton, Southampton, United Kingdom, 2 Human Development and Health, Faculty of Medicine, University of Southampton, Southampton, United Kingdom, 3 MRC Lifecourse Epidemiology Unit, Faculty of Medicine, University of Southampton, Southampton, United Kingdom, 4 Department of Women and Children's Health, School of Life Course Sciences, King's College London, London, United Kingdom, 5 NIHR Southampton Biomedical Research Centre, University of Southampton and University Hospital Southampton NHS Trust, Southampton, United Kingdom, 6 BC Childrens Hospital Research Institute, Centre for Molecular Medicine and Therapeutics, University of British Columbia, Vancouver, Canada

‡ EA and NTK are joint first authors. LP, KMG, and KAL are joint senior authors.
* kal@soton.ac.uk

**Data Availability Statement:** The data underlying the results presented in the study are available

## Abstract

### Background

Higher maternal plasma glucose (PG) concentrations, even below gestational diabetes mellitus (GDM) thresholds, are associated with adverse offspring outcomes, with DNA methylation proposed as a mediating mechanism. Here, we examined the relationships between maternal dysglycaemia at 24 to 28 weeks' gestation and DNA methylation in neonates and whether a dietary and physical activity intervention in pregnant women with obesity modified the methylation signatures associated with maternal dysglycaemia.

### Methods and findings

We investigated 557 women, recruited between 2009 and 2014 from the UK Pregnancies Better Eating and Activity Trial (UPBEAT), a randomised controlled trial (RCT), of a lifestyle intervention (low glycaemic index (GI) diet plus physical activity) in pregnant women with obesity (294 contol, 263 intervention). Between 27 and 28 weeks of pregnancy, participants had an oral glucose (75 g) tolerance test (OGTT), and GDM diagnosis was based on diagnostic criteria recommended by the International Association of Diabetes and Pregnancy Study Groups (IADPSG), with 159 women having a diagnosis of GDM. Cord blood DNA samples from the infants were interrogated for genome-wide DNA methylation levels using the Infinium Human MethylationEPIC BeadChip array. Robust regression was carried out,

from Gene Expression Omnibus under accession no. GSE141065.

**Funding:** Funding: EA, NTK, PT, JH, GCB, SLW, PS, KG, LP and KAL were funded by the Diabetes UK (16/0005454) (www.diabetes.og.uk). The UPBEAT trial was supported by the National Institute for Health Research (NIHR) (www.nihr.ac.uk) under the Programme Grants for Applied Research Programme (RP-0407-10452) and the Chief Scientist Office, Scottish Government Health Directorates (Edinburgh) (CZB/A/680) (www.cso.scot.nhs.uk). LP and SW are supported by the NIHR Biomedical Research Centre at Guys & St Thomas NHS Foundation Trust & King's College London. KMG is supported by the UK Medical Research Council (MC_UU_12011/4)(www.mrc.ukri.org), the National Institute for Health Research (NIHR Senior Investigator (NF-SI-0515-10042) and the NIHR Southampton Biomedical Research Centre) and the European Union (Erasmus+ Programme Early Nutrition eAcademy Southeast Asia-573651-EPP-1-2016-1-DE-EPPKA2-CBHE-JP) (www.ec.europa.eu). KVD is supported by the British Heart Foundation FS/17/71/32953(www.bhf.org.uk). The funders had no role in study design, data collection and analysis, decision to publish, or preparation of the manuscript.

**Competing interests:** Competing Interests:I have read the journal's policy and the authors of this manuscript have the following competing interests: KMG and GCB have received reimbursement for speaking at conferences sponsored by companies selling nutritional products. KAL and KMG are part of academic research programs that have received research funding from Abbott Nutrition, Nestec, Danone and BenevolentAI Bio Ltd. GCB has received research funding from Abbott Nutrition, Nestec and Danone and has been a scientific advisor to BASF. LP have received funding from Abbott Nutrition and Danone.The remaining authors declare no competing interests.

**Abbreviations:** AVP, arginine vasopressin; BiNGO, Biological Networks Gene Ontology; BMIQ, beta-mixture quantile; BRD2, Bromodomain Containing 2; CMMT, Centre for Molecular Medicine and Therapeutics; CpG, cytosine-phosphate-guanine; DAG, directed acyclic graph; dmCpG, differentially methylated CpG; DMR, differentially methylated region; ENCODE, Encyclopedia of DNA Elements; EWAS, epigenome-wide association study; FDR, false discovery rate; FPG, fasting plasma glucose; GCH1, GTP cyclohydrolase 1; GDM, gestational diabetes mellitus; gDNA, genomic DNA; GEM, Genotype-Environment-Methylation; GI, glycaemic index; GO, gene ontology; HAPO, Hyperglycemia and Adverse Pregnancy Outcome; HUVEC, human

adjusting for maternal age, smoking, parity, ethnicity, neonate sex, and predicted cell-type composition. Maternal GDM, fasting glucose, 1-h, and 2-h glucose concentrations following an OGTT were associated with 242, 1, 592, and 17 differentially methylated cytosine-phosphate-guanine (dmCpG) sites (false discovery rate (FDR) $\leq$ 0.05), respectively, in the infant's cord blood DNA. The most significantly GDM-associated CpG was cg03566881 located within the leucine-rich repeat-containing G-protein coupled receptor 6 (LGR6) (FDR = 0.0002). Moreover, we show that the GDM and 1-h glucose-associated methylation signatures in the cord blood of the infant appeared to be attenuated by the dietary and physical activity intervention during pregnancy; in the intervention arm, there were no GDM and two 1-h glucose-associated dmCpGs, whereas in the standard care arm, there were 41 GDM and 160 1-h glucose-associated dmCpGs. A total of 87% of the GDM and 77% of the 1-h glucose-associated dmCpGs had smaller effect sizes in the intervention compared to the standard care arm; the adjusted $r^2$ for the association of LGR6 cg03566881 with GDM was 0.317 (95% confidence interval (CI) 0.012, 0.022) in the standard care and 0.240 (95% CI 0.001, 0.015) in the intervention arm. Limitations included measurement of DNA methylation in cord blood, where the functional significance of such changes are unclear, and because of the strong collinearity between treatment modality and severity of hyperglycaemia, we cannot exclude that treatment-related differences are potential confounders.

## Conclusions

Maternal dysglycaemia was associated with significant changes in the epigenome of the infants. Moreover, we found that the epigenetic impact of a dysglycaemic prenatal maternal environment appeared to be modified by a lifestyle intervention in pregnancy. Further research will be needed to investigate possible medical implications of the findings.

## Trial registration

ISRCTN89971375.

## Author summary

### Why was this study done?

- The incidence of gestational diabetes is increasing worldwide, concurrent with a rise in obesity with children born to mothers with gestational diabetes mellitus (GDM) having a heightened risk of obesity and metabolic disease, perpetuating an intergenerational cycle of metabolic disease.

- High circulating levels of glucose in mothers with GDM have been suggested to trigger epigenetic changes (chemical modifications that affect gene activity and the amount of protein produced from them) during development of the fetus, resulting in an increased susceptibility to metabolic disease in later life.

umbilical vein endothelial cell; IADPSG, International Association of Diabetes and Pregnancy Study Groups; LGA, large-for-gestational-age; LGR6, leucine-rich repeat-containing G-protein coupled receptor 6; MAD, median absolute deviation; MCODE, Molecular Complex Detection; MDS, multidimensional scaling; meQTLs, methylation quantitative trait loci; OGTT, oral glucose tolerance test; OXT, oxytocin; PDZD8, PDZ domain containing 8 gene; PFKP, phosphofructokinase; PG, plasma glucose; PLCH1, 1-phosphatidylinositol 4,5-bisphosphate phosphodiesterase eta-1; PLEKHB1, Pleckstrin Homology Domain Containing B1; PPI, protein–protein interaction; Q–Q, quantile–quantile; RCT, randomised controlled trial; SPON1, spondin 1; STRING, Search Tool for the Retrieval of Interacting Genes/Proteins; THEMIS2, Thymocyte Selection Associated Family Member 2; TMEM210, transmembrane protein 210; TSS, transcription start site; T2D, type 2 diabetes; UPBEAT, UK Pregnancies Better Eating and Activity Trial; ZMYND8, Zinc Finger MYND-Type Containing 8.

- As little is known of the epigenetic changes induced by maternal GDM within mothers with obesity, a high-risk population for GDM, we examined relationships between DNA methylation in infants born to mothers with obesity who developed GDM and those who did not and the mother's blood glucose concentration. We then examined whether a dietary and physical activity intervention during pregnancy, designed to improve maternal glycaemia, modified the DNA methylation changes in the infant associated with maternal GDM exposure.

### What did the researchers do and find?

- Using samples from the UK Pregnancies Better Eating and Activity Trial (UPBEAT), a randomised controlled trial (RCT) of lifestyle intervention (low glycaemic index (GI) diet plus physical activity) versus standard care in pregnant women with obesity, we investigated cord blood DNA methylation levels from 557 newborn infants.

- Maternal GDM status and high circulating maternal glucose levels were associated with modest changes in DNA methylation in the infants.

- The methylation changes observed in the infant associated with maternal GDM exposure appeared to be reduced by the pregnancy lifestyle intervention.

### What do these findings mean?

- These findings suggest that the impact of high maternal circulating glucose levels on DNA methylation in the infant can be modified by a lifestyle intervention in pregnancy.

- Follow-up studies are needed to establish whether the reduction in DNA methylation changes observed in infants from mothers with GDM undertaking the lifestyle intervention is accompanied by improved health outcomes of the children in later life.

## Introduction

Maternal obesity is a major risk factor for the development of gestational diabetes mellitus (GDM) [1,2], which is defined as diabetes that develops during pregnancy. Concomitant with the rising prevalence of maternal obesity, the incidence of GDM is increasing [3–5]. Women with GDM are at high risk from pregnancy and delivery complications including infant macrosomia, neonatal hypoglycaemia, and cesarean delivery [6]. Additionally, children born to mothers with GDM have a heightened risk of obesity and metabolic disease, which may lead to an intergenerational cycle of metabolic disease [7]. Recent studies have shown that maternal GDM and dysglycaemia even below GDM thresholds are associated with adverse offspring outcomes, including increased neonatal and childhood adiposity, altered neurodevelopment, greater insulin resistance, and a lower disposition index in childhood, a key risk factor in the development of type 2 diabetes (T2D) [8–14].

One of the mechanisms by which maternal GDM increases the risk of metabolic disease in the child is suggested to be through stable modifications of the offspring's epigenome as a result of in utero exposure [15]. Epigenetic processes, which include DNA methylation,

histone modification, and noncoding RNAs, induce heritable changes in gene expression without a change in nucleotide sequence. DNA methylation can be influenced by both genotype and the environment [16]. Substantial evidence from human epidemiological studies is accruing to suggest that the in utero environment can alter the epigenome of the infant. Maternal undernutrition [17] and micronutrient status [18], maternal obesity [19,20], and socioeconomic status[21] have all been associated with changes in the methylation status of the offspring epigenome. Both candidate gene [22–24] and genome-wide studies [25–31] have reported that GDM exposure is associated with significant changes in the infant's or child's methylome, and a recent meta-analysis of 7 pregnancy cohorts identified differentially methylated regions (DMRs) associated with GDM within OR2L13 and CYP2E1 [32]. However, the majority of these studies have focussed on GDM versus no GDM, rather than the continuous relationship between maternal glucose levels and DNA methylation [33,34], and none has studied the GDM-associated signal within a high-risk population of women with obesity; thus, the relationship between the degree of maternal dysglycaemia and the contribution of maternal post-challenge/postprandial glucose excursions to the infant's methylation signature in this high-risk group remains unknown.

To date, direct evidence linking maternal GDM exposure to adverse health outcomes in the offspring through an epigenetic mechanism is lacking, as most studies have been observational [25,27,28]. Thus, there is a need for appropriately designed randomised control intervention studies that examine the causal link between dysglycaemia and induced changes in the fetal epigenome. Lifestyle interventions, particularly those designed to improve glycaemic control in women with GDM or those at risk of GDM, offer a promising strategy to improve outcomes for the mother and child. The UK Pregnancies Better Eating and Activity Trial (UPBEAT) is the largest randomised controlled trial (RCT) of a complex lifestyle intervention (low glycaemic index (GI) diet, reduced saturated fat intake, and increased physical activity) in pregnant women with obesity and has shown improvement in certain maternal and infant outcomes [35,36]. Obesity is a major risk factor for GDM[5], and 26% of the women in this study developed GDM. Although the lifestyle intervention did not prevent the 2 primary outcomes of the trial—the incidence of GDM and large-for-gestational-age (LGA) infants, it reduced maternal glycaemic load, saturated fat intake, gestational weight gain and adiposity, and improved the maternal metabolome [35,37]; in the infants, adiposity at age 6 months was reduced [36]. Here, utilising samples from the UPBEAT trial, we sought to (1) identify the DNA methylation changes in cord blood associated with maternal GDM, as well as relationships with fasting, and post-challenge 1-h and 2-h glucose concentrations; and (2) investigate whether a lifestyle intervention in pregnancy focussed on improving maternal glycaemic control modifies the methylation signature in the infant associated with maternal GDM or dysglycaemia.

## Methods

### Design of the intervention study

The present study used samples from the UPBEAT RCT (isrctn.org 89971375). Details of the study design have been reported previously [35]. Briefly, UPBEAT was a multicentre (8 inner-city National Health Service (NHS) Trust Hospitals in the United Kingdom—London (3 centres), Bradford, Glasgow, Manchester, Newcastle, and Sunderland) RCT, designed to test the effectiveness of a complex dietary and physical activity intervention in preventing GDM in women with obesity and reducing the incidence of LGA infants [35]. Women with underlying medical conditions and those prescribed metformin were excluded. The trial comprised 1,555 women ≥16 years of age, recruited between 2009 and 2014; all had a prepregnancy BMI ≥30 kg/m$^2$ and a singleton pregnancy. Participants were randomised between 15$^{+0}$ and 18$^{+6}$ weeks

gestation to either a lifestyle intervention (low GI diet, reduced saturated fat intake, and increased physical activity) or standard antenatal care. The primary outcomes of GDM and LGA did not differ significantly between the control and intervention arms, but there were improvements in some predefined maternal secondary outcomes in the intervention group, including reduced dietary glycaemic load, gestational weight gain, maternal sum-of-skinfold thicknesses, and increased physical activity [35,36], as well as an improved metabolome [37].

## Ethics statement

All aspects of the trial, including the analyses in the present study, were approved by the NHS Research Ethics Committee (UK Integrated Research Application System; reference 09/H0802/5), and all participants provided informed written consent, including women aged 16 and 17 using Fraser guidelines. This study is reported according to the "Strengthening the Reporting of Observational Studies in Epidemiology (STROBE)" guidelines (S1 Text).

## Clinical procedures

Clinical information was ascertained at time point 1 ($15^{+0}$ to $18^{+6}$ weeks' gestation). The trial protocol required an oral glucose tolerance test (OGTT) at $27^{+0}$ to $28^{+6}$ weeks', but for this study, a clinically pragmatic approach was adopted with OGTTs at $23^{+2}$ to $30^{+0}$ weeks' (mean $27^{+5}$) included. Diagnosis of GDM was according to International Association of Diabetes and Pregnancy Study Groups (IADPSG) criteria (fasting glucose $\geq$5.1 mmol/l and/or 1-h $\geq$10.0 mmol/l and/or 2-h $\geq$8.5 mmol/l in response to a 75 g oral glucose load) [38,39]. Blood was kept on ice, processed within 2 h and stored at −80˚C. Women with a positive diagnosis of GDM were treated using standard protocols, with GDM treatment predicated on the severity of the hyperglycaemia, beginning with dietary advice, followed by metformin with the addition and/or replacement with insulin if control was not achieved. There were no significant differences in the number of women receiving each treatment regime in the control and intervention arms of the study.

## DNA extraction

Genomic DNA (gDNA) was extracted from the buffy coat of umbilical cord blood samples using the QIAamp Blood DNA Mini Kit (Qiagen, UK). The quality of the gDNA was assessed by agarose gel electrophoresis, and the quantity of gDNA was checked on the NanoDrop ND-1000 (ThermoFisher Scientific, United States of America).

## Infinium Human OmniExpress genotype arrays

Single nucleotide polymorphism (SNP) genotyping was carried using Illumina Human OmniExpress 24v1.2 (Illumina, California, USA) at Edinburgh Clinical Research Facility, with imputation carried out using the EAGLE2 imputation pipeline (https://imputation.sanger.ac.uk) with the UK10K [40] and 1000 Genomes Phase 3 [41] reference panels.

## Infinium Human MethylationEPIC BeadChip array

DNA methylation using the Infinium Human MethylationEPIC BeadChip array was used to interrogate DNA methylation in 608 buffy coat samples, which included 14 technical replicates. These represented all participants with buffy coat samples available. A total of 1 μg of the gDNA was treated with Sodium Bisulfite using Zymo EZ DNA Methylation-Gold Kit (Zymo Research, Irvine, California, USA, D5007), and processing of the HumanMethylation850

(Infinium Methylation 850K; Illumina, California, USA) platform was carried out by the Centre for Molecular Medicine and Therapeutics (CMMT) (http://www.cmmt.ubc.ca).

## Infinium Human MethylationEPIC BeadChip array data processing

Infinium 850K data were processed using the Bioconductor package minfi [42] in R (version 3.4.2). Beta-mixture quantile (BMIQ) normalisation was applied to remove array biases and correct for probe design. Probes with a detection *p*-value >0.01 (*n* = 12,165) and beadcount <3 (*n* = 297) were removed from the dataset. Cytosine-phosphate-guanine (CpGs) known to cross hybridise to other locations in the genome [43] (*n* = 14,759), coinciding with SNPs [43] (*n* = 77,261), aligning to the sex chromosomes (*n* = 17,063), and non-CpG probes (*n* = 2,905) were also removed from the dataset. Probes with an absolute methylation range <10% were removed from the dataset, leaving 387,569 probes for differential methylation analysis. In total, 14 duplicate samples were included, for which the Euclidean distance was calculated, and hierarchical clustering, using complete linkage as implemented in the "hclust" function in R, was used. This grouped duplicate pairs together. The sex of samples were predicted using the "getSex" function, and 14 samples with discrepancies in assigned and predicted sex were removed from the dataset. Data were further assessed by visualisation of methylation density plots and calculation of median absolute deviation (MAD) scores. Duplicates were removed after normalisation, but before inference, the duplicate with the lowest MAD score was removed. A total of 22 samples showed aberrant methylation densities, and MAD scores lower than −5 were removed from the analysis, while 6 samples showed aberrant grouping on a multidimensional scaling (MDS) plot, separated by infant sex, and were removed from subsequent analysis. This resulted in 557 samples that were taken forward for further analysis. ComBat was applied to remove chip effects [44], and the batch-corrected methylation values used for downstream analysis. Model assumptions were assessed by visual inspection of quantile–quantile (Q–Q) plots, *p*-value histograms, and calculation of genomic inflation factor lambda (λ) values, which was below 1.2 for all analyses. DNA methylation microarray data have been deposited into Gene Expression Omnibus under accession no. GSE141065.

## Infinium Human MethylationEPIC BeadChip array data analysis

To adjust for differences in cellular heterogeneity, a reference-based prediction of the cell composition was carried out using the algorithm by Houseman and colleagues and the FlowSorted. CordBlood.450k package in R that utilises the reference for cord blood cell compositions estimated by Bakulski and colleagues [45,46]. Regression models using limma [47] were run with methylation as the outcome variable. We used the directed acyclic graph (DAG) approach [48] to select the covariates from a list compiled from the literature. The covariates selected by DAG and included in all models were maternal age and the predicted values for B cells, CD4 T cells, CD8 T cells, granulocytes, monocytes, natural killer cells, and nucleated red blood cell composition as continuous variables; and smoking (yes/no), ethnicity (white/African/Asian/ other), parity (primi-/multiparous), and neonate sex (male/female). The analysis was controlled for multiple testing with the Benjamini–Hochberg adjustment for false discovery rate (FDR). Sensitivity analyses were carried out to determine the effect of baseline maternal BMI, the intervention and gestational age at OGTT on any genome-wide methylation changes observed by including these variables individually as covariates in the regression models. Methylation QTL (mQTL) analysis was carried out using the Genotype-Environment-Methylation (GEM) package in R [16].

### Network and gene ontology enrichment

Protein–protein interaction (PPI) networks were examined using the Search Tool for the Retrieval of Interacting Genes/Proteins (STRING). Genes associated with a differentially methylated CpG (dmCpG) (FDR < 0.1) were entered into STRING and visualised in Cytoscape. The properties of the PPI network were calculated under default parameters, and only connected nodes were retained for further analysis. Large networks were further segmented using the Molecular Complex Detection (MCODE) algorithm in Cytoscape using default parameters. Enriched gene ontology (GO) terms were determined using the Biological Networks Gene Ontology tool (BiNGO) [49] to examine overrepresented GO terms. To account for the multiple CpGs per gene, *methlglm* from the methylGSA [50] package in R was used to investigate enriched GO terms, to compare to the STRING/BiNGO analysis.

### Enrichment of dmCpGs among chromatin enhancer states and histone modifications

We obtained the ChIP-sequencing (ChIP-seq) peak regions in the broadPeak format for processed Encyclopedia of DNA Elements (ENCODE) ChIP-seq experiments in human umbilical vein endothelial cells (HUVECs) as a surrogate for cord blood from the ENCODE data portal (https://www.encodeproject.org). The expanded 18-state model for HUVECs was obtained from the Epigenome Roadmap. We assessed the enrichment of dmCpGs among the chromatin states and histone modifications using Fisher exact tests, with all Illumina Infinium Human-Methylation 850 BeadChip CpGs as a background.

### Quantitative DNA methylation validation analysis by pyrosequencing

A total of 1 μg of gDNA was bisulfite-converted using the EZ DNA Methylation Gold Kit (Zymo Research) according to the manufacturer's protocol. PCR primers specific for bisulfite-converted DNA were designed using the PyroMark Assay Design Software (Qiagen). Primer sequences are shown in Table A in S1 Data. Quantitative DNA methylation analysis was carried out by pyrosequencing.

### Statistical analysis

All statistical analyses were carried out in R (version 3.4.2). Missing data were handled on a case-by-case basis, using a listwise deletion strategy. The hypergeometric distribution probability test was used to test the significance of the overlap between dmCpGs associated with GDM and different measures of maternal dysglycaemia. Fisher exact tests were used to test the enrichment of dmCpGs among the different histone modifications and chromatin enhancer states. Linear models were fitted to the pyrosequencing data adjusting for the following covariates: maternal age, smoking, ethnicity, parity, neonate sex, and the predicted values for B cells, CD4 T cells, CD8 T cells, granulocytes, monocytes, natural killer cells, and nucleated red blood cell composition. Regression diagnositics (Q–Q plots, heteroscedasticity, and Cook's distance and residuals) were checked for the top 5 dmCpGs in each epigenome-wide association study (EWAS) analysis and for all pyrosequencing regressions. The analysis plan used in the design of this study is provided as a supporting information file (S2 Text).

## Results

### Participant characteristics

Table 1 shows characteristics of the 557 participants; women from the UPBEAT study for whom cord blood samples were available. Compared to those who did not develop GDM,

**Table 1. Participant characteristics.**

| | | No GDM (*n* = 383) | GDM (*n* = 159) | All (*n* = 557[†]) |
|---|---|---|---|---|
| Neonate sex | Male (%) | 56.7 | 47.8 | 54.1 |
| Maternal ethnicity | White (%) | 75.7 | 64.2 | 72.3 |
| | Asian (%) | 5.0 | 5.7 | 5.2 |
| | Black (%) | 15.9 | 22.0 | 17.7 |
| | Other (%) | 3.4 | 8.2 | 4.8 |
| Intervention | Intervention (%) | 46.7 | 48.4 | 47.2 |
| GDM treatment | No treatment (%) | NA | 13.2 | NA |
| | Diet only (%) | | 31.4 | |
| | Metformin (%) | | 20.8 | |
| | Insulin (%) | | 14.5 | |
| | Insulin + metformin (%) | | 13.8 | |
| Parity | Primiparous (%) | 50.9 | 40.9 | 48.0 |
| Smoking | Smoker (%) | 17.2 | 15.1 | 16.6 |
| Maternal BMI (kg/m$^2$) | | 36.13 ± 4.5 | 37.27 ± 5.11 | 36.47 ± 4.74 |
| Maternal age (years) | | 30.38 ± 5.47 | 32.28 ± 5.04 | 30.95 ± 5.42 |
| Fasting insulin (mU/ml) | | 35.5 ± 42.3 (missing *n* = 7) | 40.3 ± 52.7 (missing *n* = 20) | 36.9 ± 45.5 (missing *n* = 34) |
| Fasting glucose (mmol/L) | | 4.5 ± 0.3 | 5.3 ± 0.6 | 4.8 ± 0.6 (missing *n* = 8) |
| 1-h glucose (mmol/L) | | 7.3 ± 1.41 (missing *n* = 22) | 10.1 ± 2.0 (missing *n* = 5) | 8.1 ± 2.1 (missing *n* = 36) |
| 2-h glucose (mmol/L) | | 5.5 ± 1.1 (missing *n* = 1) | 7.0 ± 1.7 (missing *n* = 1) | 6.0 ± 1.5 (missing *n* = 10) |
| *Predicted cord blood cell proportions* | | | | |
| B cell | | 0.11 ± 0.05 | 0.13 ± 0.06 | 0.11 ± 0.05 |
| CD4 T cells | | 0.14 ± 0.07 | 0.13 ± 0.08 | 0.13 ± 0.08 |
| CD8 T cells | | 0.13 ± 0.04 | 0.12 ± 0.05 | 0.12 ± 0.04 |
| Granulocytes | | 0.45 ± 0.13 | 0.43 ± 0.14 | 0.44 ± 0.13 |
| Monocytes | | 0.10 ± 0.04 | 0.11 ± 0.05 | 0.10 ± 0.04 |
| Natural killer cells | | 0.01 ± 0.03 | 0.02 ± 0.03 | 0.02 ± 0.03 |
| Nucleated red blood cells | | 0.10 ± 0.07 | 0.11 ± 0.08 | 0.10 ± 0.07 |

[†]A total of 7 women were diagnosed as having GDM at their hospital choice, although under IADPSG, definition were not classed as having GDM. Therefore, these 7 samples were removed from the GDM analysis for consistency in the statistical analysis.

GDM, gestational diabetes mellitus; IADPSG, International Association of Diabetes and Pregnancy Study Groups.

women diagnosed with GDM (*n* = 159) were older, had a higher BMI, and were more likely to be multiparous. As expected, those who developed GDM had higher fasting plasma glucose (FPG) concentrations, together with increased plasma glucose (PG) concentrations at 1-h (1-h PG) and 2-h (2-h PG) during the OGTT. Predicted cell composition of the cord blood showed that, compared to women without GDM, offspring of women who developed GDM had higher proportions of B lymphocytes and monocytes in cord blood. Comparison of the participants in this study to those without cord blood samples revealed that the mothers were on average 7 months older (*p* = 0.02), more likely to have GDM (29% versus 23%, *p* = 0.01), less likely to be multiparous (52% versus 59%, *p* = 0.01), and had a lower ethinic diversity (*p* < 0.001) (Table B in S1 Data).

## Identification of dmCpGs in cord blood associated with maternal GDM and dysglycaemia

DNA from 557 cord blood samples were interrogated for genome-wide DNA methylation levels using the Infinium Human MethylationEPIC BeadChip array. Maternal GDM was

Table 2. Summary table of EWAS analysis.

| Phenotype | dmCpGs (FDR ≤ 0.05) | | | |
|---|---|---|---|---|
| | No adjustment for intervention (*n* = 557) | Adjustment for intervention (*n* = 557) | Control arm (*n* = 294) | Intervention arm (*n* = 263) |
| GDM | 242 | 254 | 41 | 0 |
| FPG | 1 | 1 | 1 | 0 |
| 1-h PG | 592 | 704 | 160 | 2 |
| 2-h PG | 17 | 18 | 78 | 1 |

dmCpG, differentially methylated CpG; EWAS, epigenome-wide association study; FPG, fasting plasma glucose; GDM, gestational diabetes mellitus; PG, plasma glucose.

associated (Benjamini–Hochberg FDR adjusted $p < 0.05$) with altered methylation status of 242 CpG loci in cord blood (Tables 2 and 3 and Table C in S1 Data, Figs 1A, 1B and 2C), of which 7 remained significant after Bonferroni adjustment for multiple testing. The top 2 dmCpGs associated with maternal GDM were cg03566881, located within the body of the leucine-rich repeat-containing G-protein coupled receptor 6 (*LGR6*) gene (Fig 1D), and cg16536918, located within 200 bp of the transcription start site (TSS) of the arginine vasopressin (*AVP*) gene (Table 3). In total, 72.7% of the dmCpGs showed hypermethylation in those diagnosed with GDM, with an overrepresentation of dmCpGs in the OpenSea regions (Fig 1C). Adjustment for baseline maternal BMI in these analyses led to a small increase in the number of dmCpGs associated with maternal GDM status to 282 dmCpGs (adjusted *p*-value <0.05), with an 81.6% overlap ($p$-value $< 1 \times 10^{-308}$) between the GDM-associated dmCpGs with and without adjustment for maternal BMI (Table D in S1 Data). Adjustment for gestational age at OGTT resulted in 288 dmCpGs (FDR $< 0.05$) associated with GDM, with 86% of the intial 242 GDM-associated dmCpGs remaining significantly associated with GDM after gestational age adjustment.

Different pathophysiological pathways are implicated in abnormal fasting and post-challenge glucose concentrations at 1 h versus 2 h; to gain insight into whether these might induce differential effects on the infants epigenome, we analysed cord blood DNA methylation with respect to continuous measures of maternal FPG, 1-h PG, and 2-h PG concentrations post-OGTT. There was 1 dmCpG associated with maternal FPG, 592 dmCpGs associated with 1-h PG (Figs 1E, 1F and 2C), and 17 associated with 2-h PG (Tables 2 and 3 and Tables E and F in S1 Data). The dmCpG associated with FPG was cg03750061, located in the body of the PDZ domain containing 8 gene (*PDZD8*); for 1-h PG, the top 2 dmCpGs were cg0896944, located within an intergenic region on chromosome 17 (Fig 1H), and cg08960443, located within 1,500 bp of the TSS of the transmembrane protein 210 (*TMEM210*) gene (Table 3); for 2-h PG, the top 2 dmCpGs were located in the body of the proteasome 26S subunit, non-ATPase 12 (*PSMD12*) gene (cg07552638) and the GTP cyclohydrolase 1 (*GCH1*) gene (cg01899130). There was an overrepresentation of dmCpGs associated with 1-h PG in the OpenSea regions (Fig 1G), with 102 of the 1-h PG-associated dmCpGs overlapping with the GDM-associated dmCpGs ($p = 8.76 \times 10^{-222}$). There was 1 dmCpG associated with both maternal GDM and FPG and 4 dmCpGs associated with both GDM and 2-h PG (Fig 2A, Table G in S1 Data). Furthermore, there was significant overlap between the 1-h PG (58%), 2-h PG (82%), and FPG (100%) dmCpGs with the dmCpGs after additional adjustment of the regression models for gestational age at OGTT (Table H+I in S1 Data).

As there is accumulating evidence for sex differences in DNA methylation changes associated with adverse prenatal exposures [51–54], we explored differences in the methylation signatures associated with GDM exposure in male versus female infants (Table J in S1 Data). In

**Table 3. Top 10 dmCpGs with an FDR of ≤0.05 in the infant cord blood associated with maternal GDM, FPG, 1-h, and 2-h PG.**

| Probe | logFC | Average methylation | Adjusted *p*-value | Gene |
|---|---|---|---|---|
| **GDM** | | | | |
| cg03566881 | 0.0135 | 0.8733 | 0.0002 | LGR6 |
| cg16536918 | 0.0226 | 0.6758 | 0.0010 | AVP |
| cg16063640 | −0.0159 | 0.3656 | 0.0025 | |
| cg12148585 | 0.0218 | 0.4922 | 0.0056 | KLF7 |
| cg03750061 | 0.0170 | 0.6931 | 0.0056 | PDZD8 |
| cg08726900 | 0.0242 | 0.4018 | 0.0056 | ANKRD11 |
| cg08450478 | 0.0187 | 0.7011 | 0.0059 | PTGFR |
| cg11646706 | 0.0280 | 0.5653 | 0.0066 | ACOX2 |
| cg13608623 | 0.0218 | 0.6277 | 0.0066 | PBX1 |
| cg18317026 | 0.0184 | 0.8285 | 0.0066 | UNC13C |
| **Fasting glucose levels** | | | | |
| cg03750061 | 0.0148 | 0.6930 | 0.0047 | PDZD8 |
| **1-h glucose levels** | | | | |
| cg26027170 | 0.0036 | 0.3015 | 0.0041 | |
| cg08960443 | 0.0056 | 0.4756 | 0.0060 | TMEM210 |
| cg14656043 | 0.0044 | 0.3289 | 0.0060 | CREM |
| cg10020892 | 0.0045 | 0.1443 | 0.0060 | BCL9 |
| cg18422587 | 0.0046 | 0.1130 | 0.0060 | |
| cg19736654 | −0.0019 | 0.8336 | 0.0060 | TRIP10 |
| cg16581631 | 0.0015 | 0.9176 | 0.0066 | |
| cg25049210 | 0.0084 | 0.6921 | 0.0066 | |
| cg01346147 | 0.0030 | 0.8027 | 0.0066 | |
| cg06356306 | 0.0036 | 0.7917 | 0.0066 | SNAP91 |
| **2-h glucose levels** | | | | |
| cg07552638 | 0.0048 | 0.3658 | 0.0395 | PSMD12 |
| cg01899130 | 0.0105 | 0.4350 | 0.0395 | GCH1 |
| cg22407111 | 0.0057 | 0.3213 | 0.0395 | PPP2R2C |
| cg00327947 | 0.0045 | 0.3136 | 0.0395 | |
| cg07495470 | 0.0035 | 0.5169 | 0.0395 | POLDIP3 |
| cg23108535 | 0.0069 | 0.5687 | 0.0395 | MBNL1 |
| cg07302471 | 0.0055 | 0.8310 | 0.0395 | RAD54L2 |
| cg15617775 | 0.0039 | 0.1476 | 0.0395 | MAD1L1 |
| cg18724135 | 0.0022 | 0.9237 | 0.0408 | |
| cg02868516 | 0.0041 | 0.5936 | 0.0408 | RPA1 |

Regression models included maternal age, smoking, ethnicity, parity, neonate sex, and the predicted values for B cells, CD4 T cells, CD8 T cells, granulocytes, monocytes, natural killer cells, and nucleated red blood cell composition. *p*-values were adjusted for multiple testing using the Benjamini–Hochberg correction. Average methylation represents the average beta value across all samples.

dmCpG, differentially methylated CpG; FDR, false discovery rate; FPG, fasting plasma glucose; GDM, gestational diabetes mellitus; logFC, log fold change; PG, plasma glucose.

the males, 4 dmCpGs were associated with GDM and 75 dmCpGs with 1-h PG (FDR < 0.05). In females, there were no dmCpGs associated with GDM, and 112 dmCpGs associated with 1-h PG (FDR < 0.05). There was no overlap between the dmCpGs associated with 1-h PG in the males and females. Similarly, examining the top 250 dmCpGs associated with GDM and 1-h PG in the males and females, there was no overlap between the GDM-associated dmCpGs, and only 3/250 (1.2%) of the 1-h PG-associated dmCpGs in common between male and female infants.

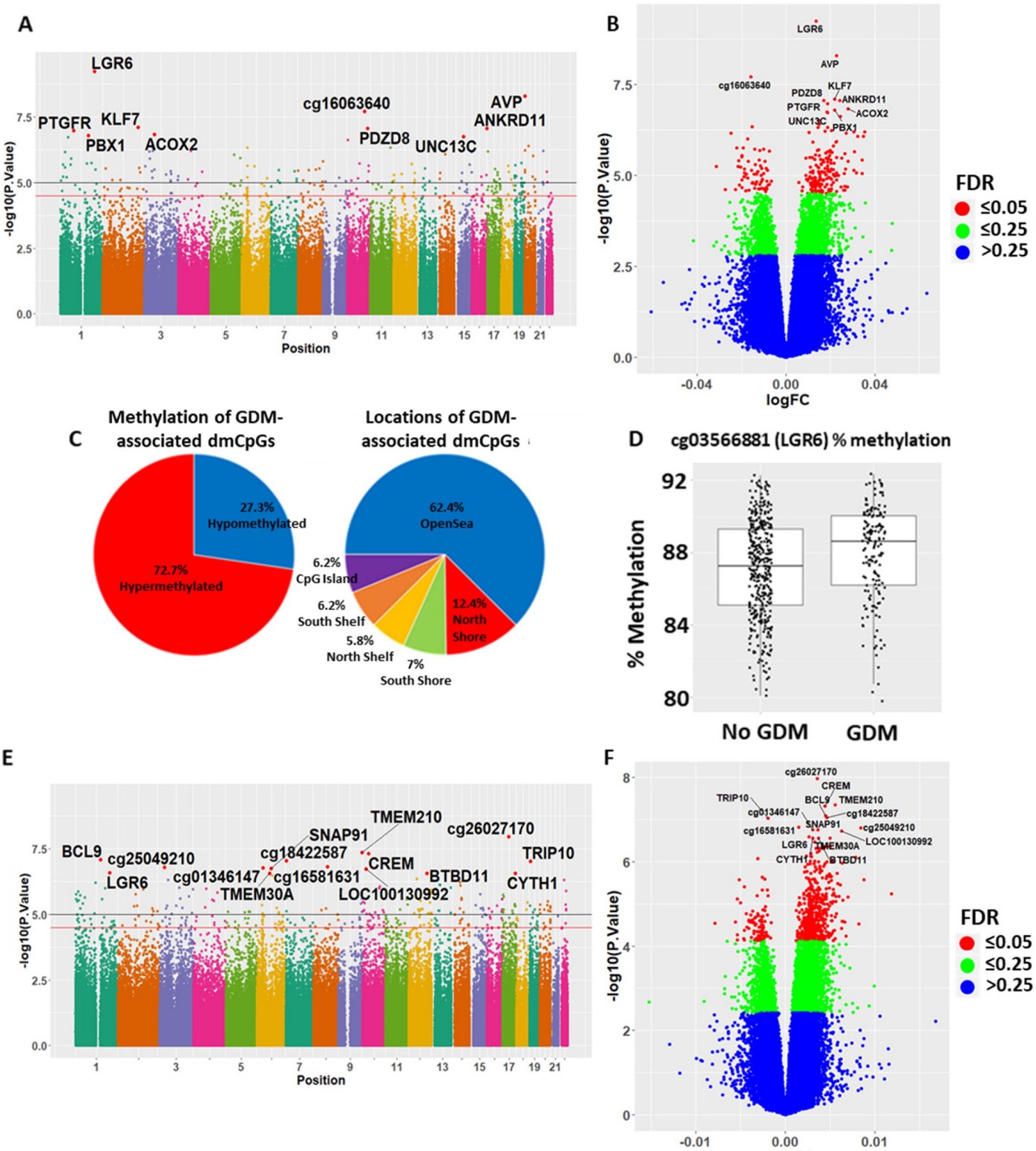

**Fig 1. Differential methylation results with respect to GDM and 1-h PG.** (A+E) Manhattan plots highlighting GDM and 1-h PG-associated dmCpGs. The black line represents $p = 1 \times 10^{-5}$, while the red line represents Bonferroni $p = 1.29 \times 10^{-7}$. (B+F) Volcano plot of the methylation results with respect to GDM and 1-h PG levels, with significant dmCpGs highlighted in red. (C+G) Pie chart showing the proportions of dmCpGs showing increased or decreased methylation (left panel) and showing the proportions of the locations relative to CpG islands (right panel). (D+H) Plot of the top dmCpG associated with GDM (cg03566881) and 1-h PG (cg26027170) as

measured on the EPIC. CpG, cytosine-phosphate-guanine; dmCpG, differentially methylated CpG; FDR, false discovery rate; GDM, gestational diabetes mellitus; logFC, log fold change; PG, plasma glucose.

## GDM-associated dmCpGs are enriched in active enhancer regions and regions overlapping H3K4 methylation/H3K27 acetylation

To investigate the functional significance of the methylation changes associated with maternal dysglycaemia, we examined the chromatin landscape surrounding the GDM and dysglycaemia-associated dmCpGs, using the ENCODE Hidden Markov Model (ChromHMM). There was a significant enrichment of GDM- and 1-h PG-associated dmCpGs overlapping active/ weak enhancers, DNase hypersensitivity sites, as well as regions of H3K4me1/3, H3K79me2, H3K27, and H3K9 acetylation, with an underrepresentation in heterochromatin and quiescent inactive regions. The 1-h PG-associated dmCpGs were also enriched in regions overlapping H3K4me2, H4K20me1, and H2A.z modification, whereas 2-h PG-associated dmCpGs were enriched only at regions overlapping H3K79me2 modification (S1 Fig).

## Maternal GDM and dysglycaemia-associated dmCpGs were enriched in networks associated with cell signalling and cell division

To determine whether sites of differential methylation were associated with specific gene networks, the 665 GDM-associated dmCpGs at an adjusted $p$-value $<0.1$ were inputted into STRING to generate a PPI network (Fig 3A). Of 665 dmCpGs, 318 were associated with a gene. There was significant biological connection between the genes ($p = 0.00161$), with 23 significant GO terms overrepresented in the PPI network (FDR $< 0.05$), with the top term being intracellular signal transduction (FDR $= 1.68 \times 10^{-4}$). To determine key modules, the network was subdivided into clusters using the MCODE algorithm, with 8 individual clusters identified. The significant GO terms associated with the clusters included multi-organism process (FDR $= 1.70 \times 10^{-3}$), chromatin remodelling complex (FDR $= 1.70 \times 10^{-7}$), chromosome, centromeric region (FDR $= 2.76 \times 10^{-7}$), antigen processing and presentation (FDR $= 1.20 \times 10^{-6}$), DNA repair (FDR $= 1.55 \times 10^{-6}$), and positive regulation of gene expression (FDR $= 5.45 \times 10^{-6}$) (Fig 3, Table 4). Network analysis for 1-h PG-associated dmCpGs (Table 4) showed an overrepresentation of GO terms associated with transcription cofactor activity (FDR $= 3.03 \times 10^{-6}$), histone deacetylase complex (FDR $= 1.45 \times 10^{-6}$), ligase activity (FDR $= 1.99 \times 10^{-6}$), integral to plasma membrane (FDR $= 3.55 \times 10^{-6}$), DNA metabolic process (FDR $= 4.61 \times 10^{-6}$), and regulation of transcription (FDR $= 2.53 \times 10^{-6}$) (Fig 3, Table 4). FPG and 2-h PG dmCpGs were not enriched for specific GO terms.

In addition, to adjust for the bias of multiple CpGs per gene, the GDM-associated dmCpGs were also inputted into *methylglm* function from the methylGSA package in R. MethylGSA analysis showed similar pathways enriched, with the GDM-associated dmCpGs showing enrichment for the GO terms cell surface receptor signalling pathway (FDR $= 0.00082$) and cytosol (FDR $= 0.00362$), while the 1-h PG-associated dmCpGs were enriched for the GO terms nuclear part (FDR $= 4.91 \times 10^{-10}$) and nucleoplasm (FDR $= 1.14 \times 10^{-9}$) (Table K in S1 Data).

## Maternal dysglycaemia is associated with differentially methylated regions in the infants' methylome

DMRs associated with maternal GDM exposure and dysglycaemia were identified using DMRcate (Tables L–N in S1 Data). GDM was associated with 47 DMRs, FPG with no DMRs, 1-h PG with 114 DMRs, and 2-h PG with 5 DMRs (Stouffer $\leq 0.05$). Of the GDM-associated

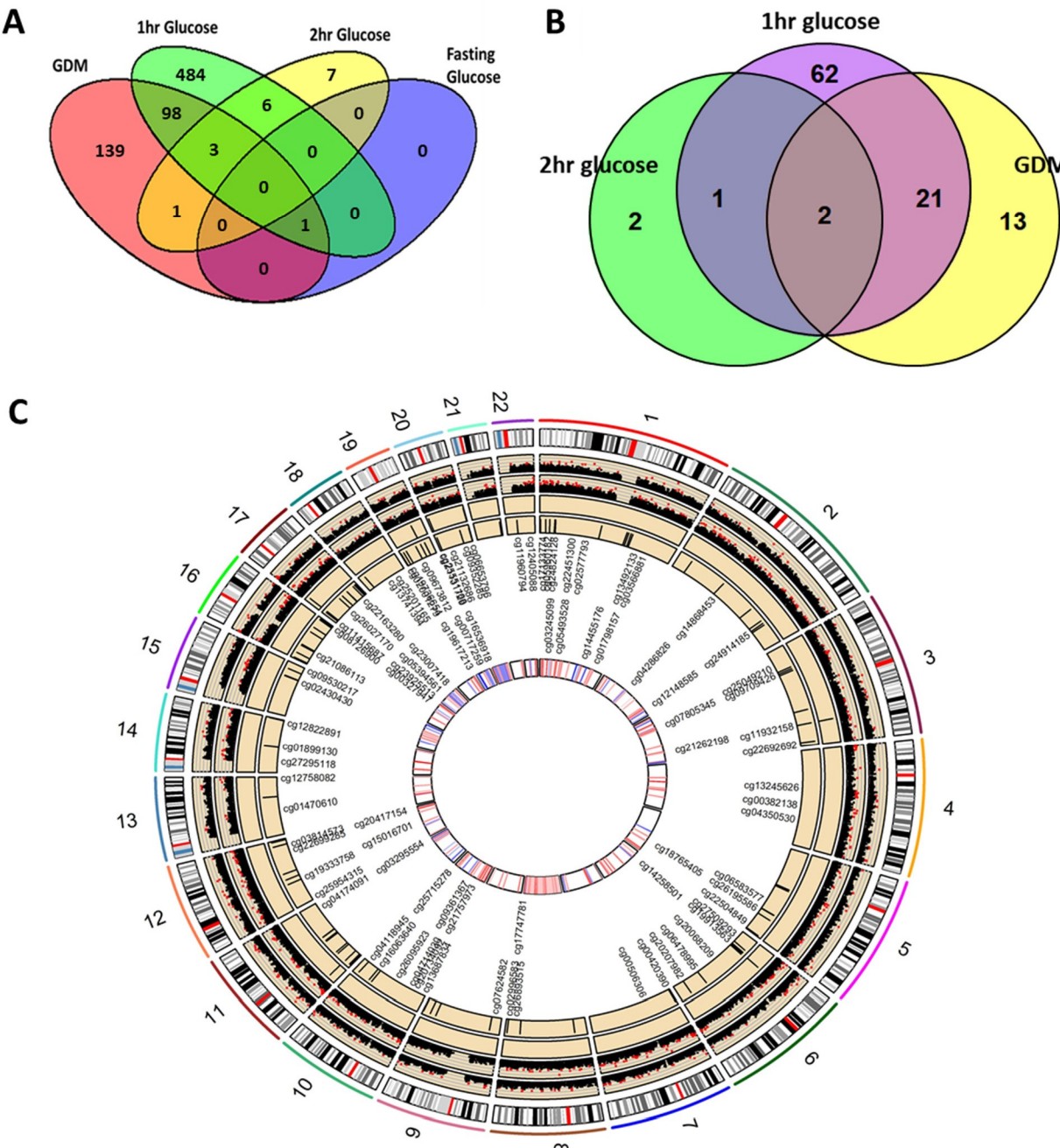

**Fig 2. Visualisation of the overlap of dmCpGs between GDM and the continuous glucose measures.** (A) Venn diagram of the overlap of the dmCpGs (FDR < 0.05) associated with maternal GDM exposure and fasting, 1-h and 2-h PG levels. (B) Venn diagram of the overlap of the DMRs associated with maternal GDM exposure, 1-h and 2-h PG. There were no DMRs associated with fasting glucose levels. (C) RCircos plot showing the distribution in the genome of the top 50 dmCpGs associated with GDM and 1-h PG levels. Track 1 (outer track) shows chromosome number, and track 2 shows the chromosome banding. Track 3 highlights the GDM-associated dmCpGs. Manhattan plots are shown for GDM (track 4) and 1-h PG (track 5) analysis, with dmCpGs FDR < 0.05 shown in red. DMRs associated with GDM (track 6) and 1-h glucose levels (track 7). Overlapping dmCpG names, between GDM and 1-h glucose, shown on the inside, with the innermost track highlighting whether the association between the dmCpGs and GDM are positive (red) or negative (blue). dmCpG, differentially methylated CpG; DMR, differentially methylated region; FDR, false discovery rate; GDM, gestational diabetes mellitus; PG, plasma glucose.

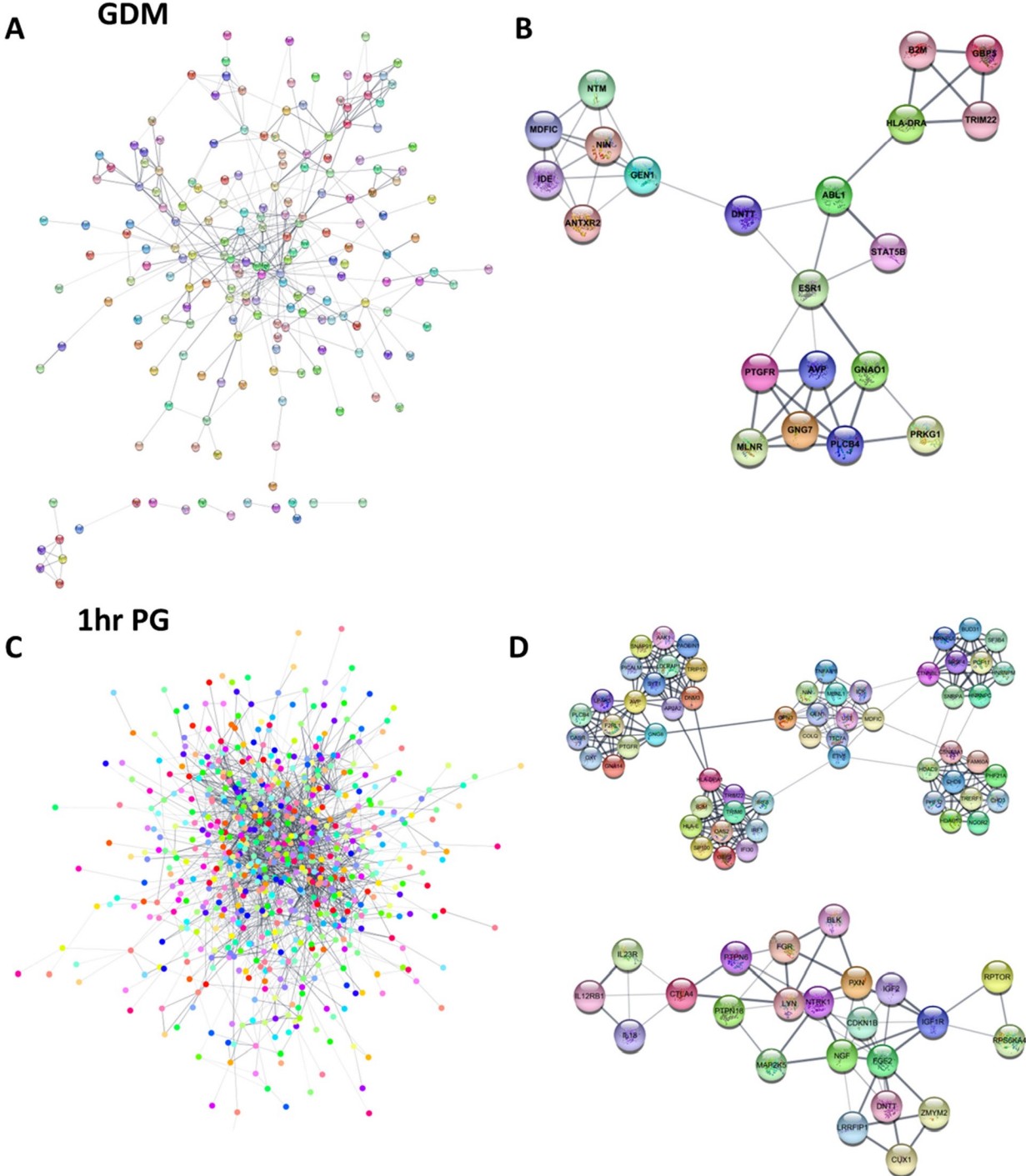

**Fig 3. PPI networks and clusters.** Networks associated with (A) GDM and (C) glucose levels 1-h post-OGTT and the top 2 modules associated with (B) GDM dmCpGs and (D) 1-h PG levels dmCpGs. dmCpG, differentially methylated CpG; GDM, gestational diabetes mellitus; OGTT, oral glucose tolerance test; PG, plasma glucose; PPI, protein–protein interaction.

DMRs, 20 of the 47 DMRs also had significant dmCpGs associated with GDM overlapping the reported DMR location (Table O in S1 Data).

**Table 4. GO terms associated with GDM and 1-h PG clusters.**

| GO ID | FDR | Description | Genes in test set |
|---|---|---|---|
| **GDM cluster 1** | | | |
| **51704** | 1.70E-03 | multi-organism process | STAT5B\|PTGFR\|MDFIC\|IDE\|AVP\|ESR1\|B2M\|TRIM22 |
| **60089** | 2.67E-03 | molecular transducer activity | GNAO1\|STAT5B\|PTGFR\|PLCB4\|GNG7\|MLNR\|HLA-DRA\|IDE\|AVP\|ANTXR2\|ESR1 |
| **GDM cluster 2** | | | |
| **775** | 2.76E-07 | chromosome, centromeric region | RCC2\|MAD1L1\|NSL1\|CLASP2 |
| **280** | 7.10E-07 | nuclear division | RCC2\|MAD1L1\|NSL1\|CLASP2 |
| **GDM cluster 3** | | | |
| **19882** | 1.20E-06 | antigen processing and presentation | HLA-DMA\|TAP2\|TAP1\|HLA-DRA\|B2M |
| **2474** | 9.08E-05 | antigen processing and presentation of peptide antigen via MHC class I | TAP2\|TAP1\|B2M |
| **GDM cluster 4** | | | |
| **10628** | 5.45E-06 | positive regulation of gene expression | EBF1\|PRDM1\|TCF3\|FGF2\|ETS1\|FOXO1 |
| **10604** | 3.92E-05 | positive regulation of macromolecule metabolic process | EBF1\|PRDM1\|TCF3\|FGF2\|ETS1\|FOXO1 |
| **GDM cluster 5** | | | |
| **16043** | 2.04E-02 | cellular component organisation | SYK\|FNBP1\|TRIP10\|BAIAP2\|VAV2 |
| **30031** | 2.04E-02 | cell projection assembly | BAIAP2\|VAV2 |
| **GDM cluster 6** | | | |
| **6281** | 1.55E-06 | DNA repair | GEN1\|RAD51B\|MGMT\|MSH3\|MLH3 |
| **6974** | 3.12E-06 | response to DNA damage stimulus | GEN1\|RAD51B\|MGMT\|MSH3\|MLH3 |
| **GDM cluster 7** | | | |
| **16585** | 1.28E-07 | chromatin remodelling complex | NCOR2\|HDAC4\|SMARCD3\|ESR1 |
| **8134** | 1.70E-04 | transcription factor binding | NCOR2\|HDAC4\|SMARCD3\|ESR1 |
| **1-h glucose cluster 1** | | | |
| **118** | 1.45E-06 | histone deacetylase complex | NCOR2\|HDAC10\|CSNK2A1\|CHD3\|HDAC9\|PHF21A |
| **16585** | 6.28E-05 | chromatin remodelling complex | NCOR2\|HDAC10\|CSNK2A1\|CHD3\|HDAC9\|PHF21A |
| **1-h glucose cluster 2** | | | |
| **16563** | 2.69E-06 | transcription activator activity | NR5A1\|NCOA2\|RBM14\|SMARCD3\|MED24\|MAML2\|TBL1XR1\|MAML3\|MED26\|FOXO1 |
| **3713** | 2.69E-06 | transcription coactivator activity | NR5A1\|NCOA2\|RBM14\|SMARCD3\|MED24\|MAML2\|MAML3\|MED26 |
| **1-h glucose cluster 4** | | | |
| **6259** | 4.61E-04 | DNA metabolic process | GEN1\|RAD51B\|PAPD7\|ERCC4\|RPA1\|DNTT\|NUP98\|MLH3 |
| **6996** | 7.52E-04 | organelle organisation | ACTA1\|PAPD7\|ERCC4\|LMNA\|H3F3A\|RPA1\|EHMT1\|PTK2B\|NUP98\|JARID2\|MLH3 |
| **1-h glucose cluster 5** | | | |
| **5887** | 3.55E-05 | integral to plasma membrane | CHRNB4\|ADCY9\|ADORA3\|CCRL2\|CTLA4\|STOM\|PTH2R\|ADRB1\|PTH1R\|IL12RB1\|CCR5 |
| **31226** | 3.55E-05 | intrinsic to plasma membrane | CHRNB4\|ADCY9\|ADORA3\|CCRL2\|CTLA4\|STOM\|PTH2R\|ADRB1\|PTH1R\|IL12RB1\|CCR5 |
| **1-h glucose cluster 7** | | | |
| **3712** | 3.03E-07 | transcription cofactor activity | NCOA2\|RBM14\|MED24\|MAML2\|TBL1XR1\|MAML3\|MED26 |
| **3713** | 3.22E-07 | transcription coactivator activity | NCOA2\|RBM14\|MED24\|MAML2\|MAML3\|MED26 |
| **1-h glucose cluster 9** | | | |

(*Continued*)

**Table 4.** (Continued)

| GO ID | FDR | Description | Genes in test set |
|---|---|---|---|
| 16874 | 1.99E-05 | ligase activity | ZNRF1\|HERC3\|FBXW11\|SIAH2\|FBXO11 |
| 16881 | 5.96E-05 | acid-amino acid ligase activity | HERC3\|FBXW11\|SIAH2\|FBXO11 |
| **1-h glucose cluster 11** | | | |
| 6355 | 2.53E-03 | regulation of transcription, DNA dependent | NFIB\|DNMT3A\|ABL1\|HMGA2\|ESR1 |
| 51252 | 2.53E-03 | regulation of RNA metabolic process | NFIB\|DNMT3A\|ABL1\|HMGA2\|ESR1 |

For the GDM cluster 8 and 1-h PG clusters 3/6/8/10, there was no significant enrichment of GO terms.

FDR, false discovery rate; GDM, gestational diabetes mellitus; GO, gene ontology; MHC, major histocompatibility complex; PG, plasma glucose.

DMRs within Pleckstrin Homology Domain Containing B1 (*PLEKHB1*) and Zinc Finger MYND-Type Containing 8 (*ZMYND8*) were associated with GDM, 1-h PG, and 2-h PG levels, while DMRs within the Bromodomain Containing 2 (*BRD2*) and *TMEM210* genes were associated with both 1-h and 2-h PG levels; 26 DMRs were associated with both GDM and 1-h PG levels, including Thymocyte Selection Associated Family Member 2 (*THEMIS2*), 1-phosphatidylinositol 4,5-bisphosphate phosphodiesterase eta-1 (*PLCH1*), *AVP*, spondin 1 (*SPON1*), and oxytocin (*OXT*) genes (Table P in S1 Data, Fig 2B).

## Validation

Sodium bisulfite pyrosequencing was used to validate *AVP* cg16536918 (Fig 4A–4D) and *LGR6* cg03566881, the top hits associated with GDM; both CpGs were also associated with 1-h PG (Table 5). Consistent with the findings from the array, pyrosequencing showed that methylation of *AVP* cg16536918 and *LGR6* cg03566881 were significantly associated with GDM exposure (*AVP* cg16536918, $r^2 = 0.334$, $p = 0.012$; *LGR6* cg03566881, $r^2 = 0.263$, $p = 0.049$) and 1-h PG (*AVP* cg16536918, $r^2 = 0.343$, $p = 0.0034$; LGR6 cg03566881, $r^2 = 0.272$, $p = 0.019$). For both CpGs, there were strong correlations between the methylation levels of the CpGs on the EPIC array and the pyrosequencer (*AVP* cg16536918: $\rho = 0.7944$, $p < 2.2 \times 10^{-16}$; *LGR6* cg03566881: $\rho = 0.5093$, $p < 2.2 \times 10^{-16}$) (Fig 4A–4D).

In addition, a CpG within the *BRD2* DMR (Fig 4E–4H), which was associated with both 1-h and 2-h PG levels, was validated; cg26953232 methylation was associated with 1-h PG ($r^2 = 0.273$, $p = 0.013$) and strongly correlated with the methylation levels measured on the EPIC array ($\rho = 0.6785$, $p < 2.2 \times 10^{-16}$). In total, 3 flanking CpGs not covered on the EPIC array were also measured on the pyrosequencer, and all 3 were associated with 1-h PG (CpG2 $r^2 = 0.135$, $p = 0.047$; CpG3 $r^2 = 0.204$, $p = 0.029$; CpG4 $r^2 = 0.134$, $p = 0.037$) (Fig 4E and 4F). All the assays analysed by pyrosequencing validated the results obtained from the array.

## The effect of the lifestyle intervention on the methylation signature associated with maternal dysglycaemia and GDM

The UPBEAT dietary and physical activity intervention did not reduce the incidence of GDM or the number of LGA infants, but improvements in maternal diet, physical activity, and a decrease in skinfold thickness and gestational weight gain [35] and improved metabolome [37] was observed in the mothers, while in the infants, there was a reduction in subscapular skinfold thickness z-score at age 6 months [36]. Here, we found that the intervention itself was not associated with significant changes in DNA methylation (FDR ≤ 0.05) in cord blood, but examination of the modifying effects of the intervention on the methylation signatures

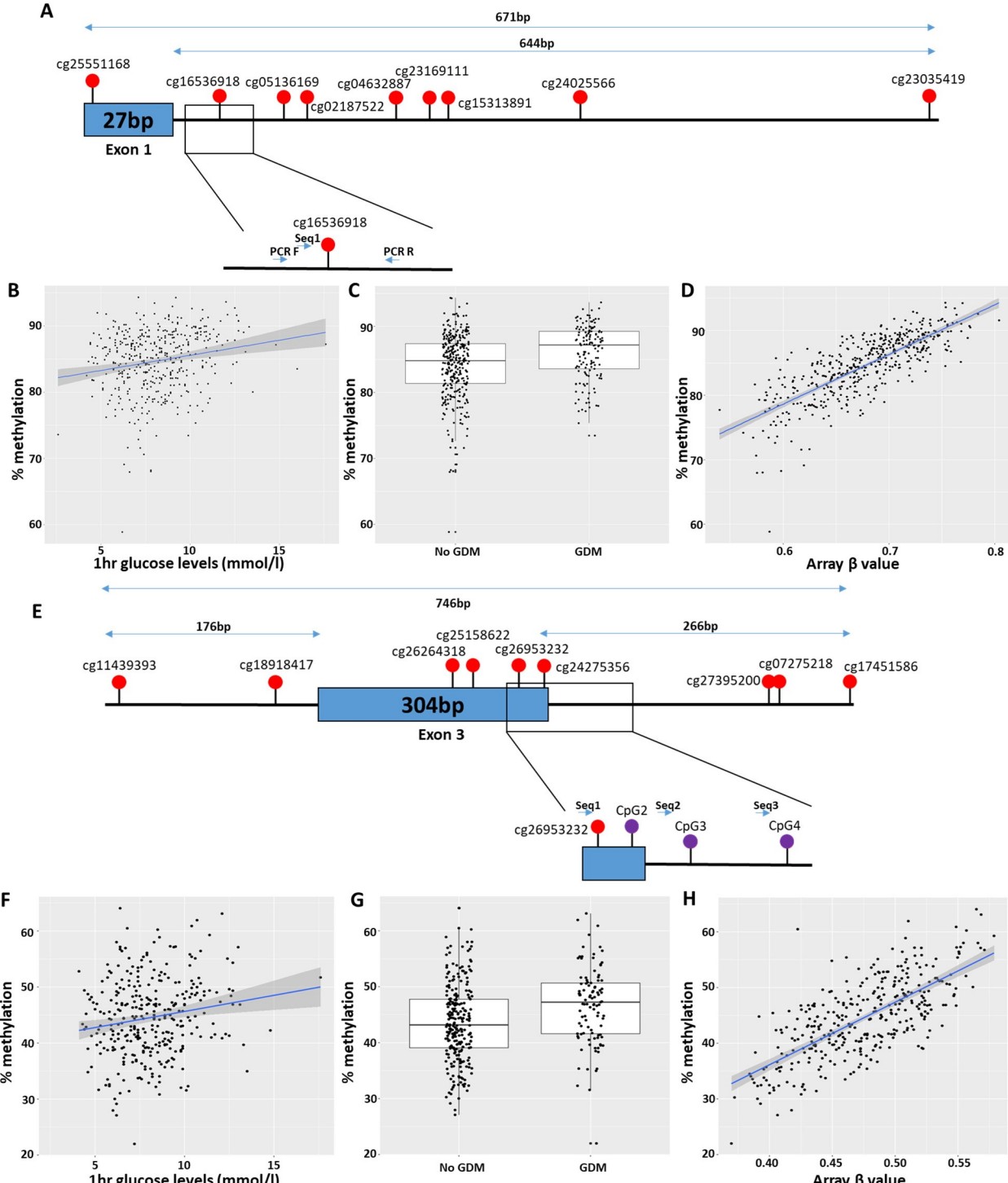

**Fig 4. Validation results of the array using bisulphite pyrosequencing.** (A–D) Validation of cg16536918, in the AVP gene, by pyrosequencing, showing (A) the location of the CpG in the AVP DMR found associated with GDM, (B) the significant association with 1-h PG levels, (C) GDM status, and (D) the correlation between methylation levels measured on the pyrosequencer and beta values on the array. (E–H) Validation of cg26953232, a CpG in the DMR, found associated with 1-h PG levels, showing (E) the location of the CpG in the genome, (F) the significant association with 1-h PG, (G) GDM status, and (H) the correlation between methylation levels measured on the pyrosequencer and beta values on the array. (A+E) Diagrams of the dmCpGs and the location of the (A) AVP and (E) BRD2 pyrosequencer assay. Red circles indicate CpGs on the array identified as part of the DMR. Purple circles indicate extra CpGs measured on the pyrosequencer, with the location of the 3 sequencing primers used (Seq1–3). AVP, arginine vasopressin; BRD2, Bromodomain Containing 2; CpG, cytosine-phosphate-guanine; dmCpG, differentially methylated CpG; DMR, differentially methylated region; GDM, gestational diabetes mellitus; PCR, polymerase chain reaction; PG, plasma glucose.

**Table 5. Validation of dmCpGs by pyrosequencing.**

| CpG | Gene | Position | Array analysis | | | | | Pyrosequencing analysis (with respect to 1-h glucose levels) | | | | | Array vs pyrosequencing correlation | | |
|---|---|---|---|---|---|---|---|---|---|---|---|---|---|---|---|
| | | | n | beta | p-value | 95% CI | | n | beta | p-value | 95% CI | | n | rho | p-value |
| cg03566881 | LGR6 | chr1:202210983 | 521 | 0.0026 | 3.17E-07 | 0.0016 | 0.0036 | 341 | 0.1976 | 0.03049 | 0.0187 | 0.3765 | 362 | 0.5093 | <2.2E-16 |
| cg16536918 | AVP | chr10:3065403 | 521 | 0.0043 | 2.80E-06 | 0.0025 | 0.0061 | 443 | 0.3986 | 3.39E-05 | 0.2116 | 0.5856 | 458 | 0.7944 | <2.2E-16 |
| cg26953232 | BRD2 DMR | chr6:32942495 | 521 | 0.0031 | 3.27E-04 | 0.0014 | 0.0048 | 331 | 0.4231 | 0.01238 | 0.0922 | 0.7541 | 345 | 0.6785 | <2.2E-16 |
| CpG2 | | chr6:32942508 | - | - | - | - | - | 316 | 0.4386 | 0.04738 | 0.0051 | 0.8721 | - | - | - |
| CpG3 | | chr6:32942591 | - | - | - | - | - | 321 | 0.3618 | 0.02907 | 0.0372 | 0.6865 | - | - | - |
| CpG4 | | chr6:32942628 | - | - | - | - | - | 299 | 0.4343 | 0.03710 | 0.0261 | 0.8424 | - | - | - |
| **CpG** | **Gene** | **Position** | **Array analysis** | | | | | **Pyrosequencing analysis (with respect to GDM)** | | | | | | | |
| | | | n | beta | p-value | 95% CI | | n | beta | p-value | 95% CI | | | | |
| cg03566881 | LGR6 | chr1:202210983 | 383 vs 159 | 0.0128 | 2.12E-08 | 0.0084 | 0.0173 | 263 vs 97 | 0.7141 | 0.04914 | 0.0027 | 1.4256 | | | |
| cg16536918 | AVP | chr10:3065403 | 383 vs 159 | 0.0215 | 1.57E-07 | 0.0136 | 0.0295 | 259 vs 96 | 1.2632 | 0.01178 | 0.2820 | 2.2443 | | | |

CI, confidence interval; CpG, cytosine-phosphate-guanine; dmCpG, differentially methylated CpG; GDM, gestational diabetes mellitus.

associated with maternal dysglycaemia showed an increase in the number of dmCpGs associated with maternal 1-h PG concentrations after adjustment for intervention compared to the unadjusted analysis (Table 2); 704 dmCpGs were associated with 1-h PG (Table Q in S1 Data) in the intervention adjusted analysis compared to 592 in the unadjusted analysis (Table 2), with a significant overlap between the dmCpGs (82%, $p < 1 \times 10^{-308}$). There was also a small increase in the number of dmCpGs associated with maternal GDM (Table R in S1 Data) and 2-h PG after adjustment for intervention but no effect of adjusting for intervention in the analysis between maternal FPG and cord blood DNA methylation. The increase in number of dmCpGs associated particularly with 1-h PG levels after adjustment for intervention may imply a better model fit after adjustment, suggesting that the intervention might modify the 1-h PG-associated methylation signature in the infant. To investigate this further, the effects of maternal GDM exposure and dysglycaemia on cord blood DNA methylation were analysed in the standard care and intervention arms of the UPBEAT trial separately (Table 2). Here, we found there were marked differences in the number of dmCpGs associated with maternal GDM and dysglycaemia in the 2 arms of the trial (Table 2), even though there were no differences in the number of women with or without GDM, or in fasting, 1-h or 2-h PG concentrations between the standard care and intervention arms (Table 6, S2 Fig). There were 160 dmCpGs (FDR ≤ 0.05) in cord blood associated with maternal 1-h PG concentrations in the standard care arm and only 2 dmCpGs associated with 1-h PG in the intervention arm; 41 dmCpGs were associated with GDM in the standard care arm but no dmCpGs associated with GDM in the intervention arm; 78 dmCpGs associated with 2-h PG in the standard care arm and 1 in the intervention arm, while for FPG, there was 1 dmCpG in the standard care arm and no dmCpGs in the intervention arm (Tables S and T in S1 Data). Furthermore, comparison of the effect sizes for the 41 dmCpGs associated with GDM in the standard care and intervention arms of the study showed that for 87.8% of these CpGs, there were reductions in the effect size in the intervention arm compared to the standard care arm (Table U in S1 Data). Similarly, analysis of the 160 dmCpGs associated with 1-h PG showed that the effect sizes were reduced in the intervention arm compared to the standard care arm for 77.5% of the dmCpG sites. For cg03566881(LGR6) and cg16536918 (AVP), 2 of the top hits associated with GDM,

**Table 6. Cohort characteristics separated by intervention group.**

| | | Control (*n* = 294) | Intervention (*n* = 263) |
|---|---|---|---|
| Neonate sex | Male (%) | 53.7 | 54.4 |
| Maternal ethnicity | White (%) | 71.8 | 71.5 |
| | Asian (%) | 4.4 | 5.7 |
| | Black (%) | 18.7 | 17.9 |
| | Other (%) | 5.1 | 4.9 |
| GDM | GDM (%) | 38.7 | 30.1 |
| GDM treatment (% of GDM participants) | No treatment (%) | 15.7 | 12.3 |
| | Diet only (%) | 36.8 | 30.1 |
| | Metformin (%) | 18.4 | 26.0 |
| | Insulin (%) | 11.8 | 19.2 |
| | Insulin + metformin (%) | 17.1 | 12.3 |
| Parity | Primiparous (%) | 50.9 | 40.9 |
| Maternal smoking | Smoker (%) | 17.0 | 16.0 |
| Maternal BMI (kg/m$^2$) | | 36.33 ± 4.42 | 36.62 ± 5.09 |
| Maternal age (years) | | 30.91 ± 5.46 | 30.99 ± 5.38 |
| Fasting glucose (mmol/L) | | 4.7 ± 0.1 (missing *n* = 3) | 4.8 ± 0.2 (missing *n* = 5) |
| 1-h glucose (mmol/L) | | 8.2 ± 0.3 (missing *n* = 21) | 8.0 ± 0.2 (missing *n* = 15) |
| 2-h glucose (mmol/L) | | 5.9 ± 0.1 (missing *n* = 4) | 6.0 ± 0.2 (missing *n* = 6) |
| *Predicted cord blood cell proportions* | | | |
| B cell | | 0.11 ± 0.05 | 0.12 ± 0.05 |
| CD4 T cells | | 0.13 ± 0.08 | 0.13 ± 0.08 |
| CD8 T cells | | 0.12 ± 0.05 | 0.12 ± 0.04 |
| Granulocytes | | 0.44 ± 0.13 | 0.45 ± 0.13 |
| Monocytes | | 0.11 ± 0.04 | 0.10 ± 0.04 |
| Natural killer cells | | 0.02 ± 0.03 | 0.02 ± 0.03 |
| Nucleated red blood cells | | 0.11 ± 0.08 | 0.10 ± 0.07 |

GDM, gestational diabetes mellitus.

the adjusted r$^2$ in the analysis stratified by intervention showed a decrease in effect size from 0.317 to 0.240 and from 0.292 to 0.258, respectively, in the intervention arm compared to the standard care arm, with similar decreases in the 1-h PG analysis.

## Influence of genetic variation on the dysglycaemia-associated dmCpGs

As DNA methylation can be driven by genotype [30,31], we investigated the potential influence of genetic variants on the 160 1-h PG-associated dmCpGs in the standard care arm of the UPBEAT RCT by carrying out a genome-wide mQTL screen using the *GEM* package[16]. For the majority of dmCpGs associated with 1-h PG, there was no significant association with genotype, but significant mQTLs associations between 666 genetic variants and 18 of the 160 dmCpGs (Fig 5, Table V in S1 Data) was observed. Of these, the majority (613) of the mQTLs were *trans*-mQTLs, with 53 *cis*-mQTLs. Comparison with the ARIES mQTL dataset in cord blood, generated using the Illumina 450K methylation array, revealed only 6 of the 18 dmCpGs, identified as being associated with genetic variation in this study, were found on the Illumina 450K array; none had previously been reported as mQTLs, although cg12053291 and cg06388350 showed weak mQTLs in the ARIES dataset (FDR < 0.1). The remaining 12 of the 18 dmCpGs had not previously been interrogated in cord blood.

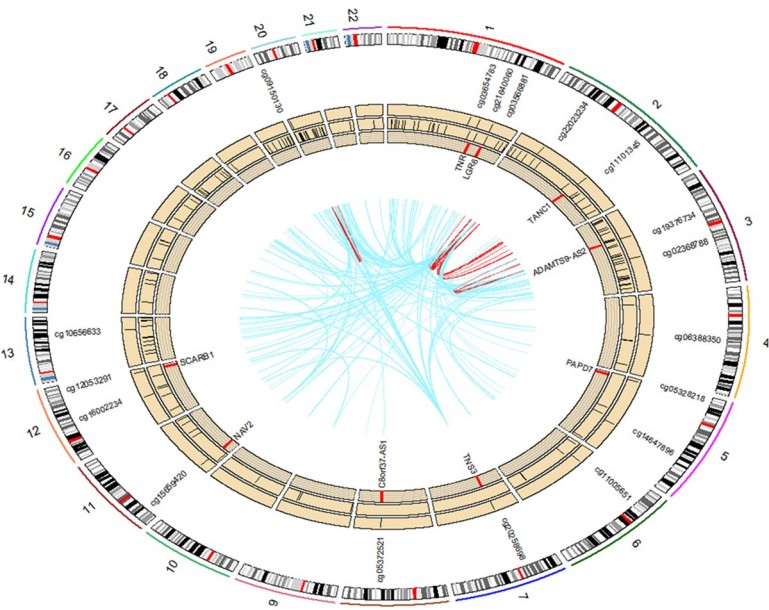

**Fig 5. RCircos plot showing the influence of genetic variation on the dysglycaemia-associated dmCpGs distribution.** The outer track (track 1) indicates chromosome number with track 2 showing the chromosome banding. Track 3 names the dmCpGs that show a significant influence of genetics and marked on track 4. Track 5 indicates the SNPs that are significantly associated with the dmCpGs. The genes that are associated with the dmCpGs are highlighted in red on track 6 and named on the inner side of track 5. The inner track shows the significant links between the CpGs and the SNPs. Blue links indicate *trans*-CpG–SNP associations, while red links indicate *cis*-CpG–SNP associations. CpG, cytosine-phosphate-guanine; dmCpG, differentially methylated CpG; SNP, single nucleotide polymorphism.

## Discussion

There is increasing evidence that maternal dysglycaemia has adverse effects on the health of the offspring, predisposing the infant to developing obesity and metabolic disease in later life [11,13,55]. The mechanisms by which the maternal environment may induce such long-term effects on the offspring have been suggested to involve the altered epigenetic regulation of genes [56]. Here, we show that maternal GDM, FPG, 1-h, and 2-h PG levels were associated with significant changes in the cord blood DNA methylome, with dmCpG loci being primarily associated with genes involved in cell signalling and transcriptional regulation. Moreover, we show that a maternal lifestyle intervention in pregnancy appeared to attenuated the GDM, 1-h, and 2-h PG-associated methylation changes in cord blood, demonstrating that the epigenetic impact of a dysglycaemic prenatal maternal environment can be modified by a maternal life-style intervention in pregnancy.

The methylation signature associated with maternal GDM status overlapped at both the dmCpG and DMR level with the methylation changes associated with maternal FPG, 1-h and 2-h PG, the 3 definitional components of GDM. The majority of the dmCpGs associated with GDM or 1-h PG were hypermethylated and enriched among enhancers, DNaseI hypersensi-tive sites, and sites of K4me1/3 and K79me2—activating histone promoter-associated marks [57], suggesting that maternal dysglycaemia is associated with increased gene silencing at sites of active gene transcription or regulation. Moreover, as the GDM and 1-h PG-associated dmCpGs were enriched in pathways associated with cell signalling and chromatin remodel-ling, this may suggest dysregulation of such pathways in GDM infants. DMRs within *ZMYND8* and *PLEKHB1* were common to GDM, FPG, 1-h, and 2-h PG; *ZMYND8* is an

epigenetic reader that provides a structural template for histone peptide recognition [58] and is known to be involved in coordinating gene expression programs associated with cell proliferation, migration, and DNA repair. Interestingly, *BRD2*, which contained a DMR associated with both 1-h and 2-h PG, is known to interact with *ZMYND8* [59] and was previously reported by Houde and colleagues to be associated with maternal hyperglycaemia [60]. *BRD2* has been reported to play a major role in metabolism [61]; it is highly expressed in pancreatic "beta" cells, where it inhibits "beta" cell mitosis and insulin transcription. *BRD2* also plays a key role in B lymphocyte cell expansion [62]. Interestingly, B lymphocyte cell number was increased in the cord blood of infants born to GDM mothers, but whether this is related to altered *BRD2* methylation in these cells requires further investigation.

There were differences in the number and location of dmCpGs associated with maternal FPG, 1-h, and 2-h PG concentrations. The strongest signal was observed with maternal 1-h PG levels, with weaker signals associated with FPG and 2-h PG. This is unlikely to reflect a difference in the number of individuals with FPG, 1-h, or 2-h PG levels above the IADPSG GDM cutoffs, as in this study there were more women with a FPG $\geq$5.1 mmol/l than women with a 1-h PG $\geq$10.0 mmol/l. Rather, these differences in methylation may arise from differences in the aetiology of the dysglycaemia; higher maternal 1-h PG levels concentrations reflect impaired insulin secretion [63–65], and our data suggest this may be a stronger driver of epigenetic change in the fetus. In the Hyperglycemia and Adverse Pregnancy Outcome (HAPO) cohort, Lowe and colleagues also reported that maternal 1-h PG levels, but not maternal FPG levels, were positively associated with the child's FPG, 1-h, and 2-h PG levels at age 10 years, and negatively associated with their disposition index [12]. As children with low disposition index are most likely to progress to T2D [9,66,67], this suggests the importance of maternal 1-h PG levels as a long-term risk factor and is consistent with the more pronounced epigenetic changes seen in the infant at birth associated with this measure of maternal dysglycaemia.

The neonatal methylation signature associated with maternal GDM/dysglycaemia differed between the sexes. A growing literature suggests sexual dimorphism in the association between maternal GDM and childhood outcomes [7–14], with a report from the OBEGEST cohort showing that exposure to GDM is a risk factor for childhood overweight in boys but not in girls [68]. Here, we found marked differences in the cord blood DNA methylation patterns associated with maternal GDM/dysglycaemia between male and female infants. Follow-up studies to investigate sex differences in the anthropometric and metabolic characteristics of the older children are therefore indicated, together with further studies to determine whether the dysglycaemia-associated dmCpGs in males and female infants are associated with specific metabolic characteristics of the children.

The lifestyle intervention did not induce significant changes in the cord blood methylome but appeared to be associated with marked attenuation of the methylation signature linked to maternal GDM, 1-h, and 2-h PG concentrations. This may reflect recommendations for decreased consumption of high GI foods in the intervention arm, leading to lower glycaemic excursions in those individuals metabolically susceptible to high early postprandial glucose concentrations; although not influencing the OGTT result, over the time frame of the intervention the low GI diet may be sufficient to lessen the impact on the offspring's methylome. Some human studies and experiments in animal models have suggested that the epigenome is most susceptible to environmental factors during the periconceptional period [69,70]. For example, in the Dutch Hunger Winter cohort, offspring DNA methylation changes were only observed in those exposed to famine during the periconceptional period rather than during the second or third trimester [71]. However, the fact that the UPBEAT lifestyle intervention appeared to attenuate the dysglceamia associated methylation signature in cord blood suggests that there is a plasticity in the cord blood methylome in the later stages of pregnancy, and

targeting interventions at the second trimester of pregnancy may have beneficial effects on off-spring outcomes.

DNA methylation can be influenced by the genotype of the individual, with sequence variation at specific loci resulting in different patterns of DNA methylation [72]. These sites are called methylation quantitative trait loci (meQTLs) and contribute to interindividual differences in DNA methylation and differential response to environmental factors. Here, we found that 11% of the 1-h PG-associated CpGs were influenced by genotype; of these, the majority were *trans*-meQTLs rather than *cis*-meQTLs. Whether the SNPs affecting these *trans*-meQTLs represent sequence variations within genes associated with the methylation machinery or in regions spatially linked to the dmCpGs is unknown. Given that the majority of 1-h PG-associated dmCpGs were located within intergenic regions and linked to enhancer regions, an understanding of the spatial 3D relationship between the meQTLs and associated SNPs may provide novel insights into the regulation of these dmCpGs.

There are several strengths to this study. Firstly, this is the largest study to date where DNA methylation changes have been examined with respect to maternal GDM exposure in obese women, a high-risk group, and all pregnancies with available cord blood were included rather than a case control design. Previous genome-wide studies investigating methylation changes associated with GDM exposure have also identified GDM-associated dmCpGs; Finer and colleagues identified 1,485 dmCpGs associated with GDM (adjusted *p*-value ≤0.05) in the cord blood of infants from mothers with GDM (*n* = 25) or without (*n* = 21) [25]. Hjort and colleagues compared DNA methylation in 9- to 16-year-old children born to mothers with (*n* = 93) or without (*n* = 95) GDM, identifying 76 GDM-associated dmCpGs [28]. While there was no overlap between the dmCpGs identified in this study with those reported by Hjort and colleagues, phosphofructokinase (*PFKP*) and pre-mRNA-splicing factor (*SYF1*) associated with GDM in this study were also identified by Finer and colleagues [25]. The lack of major overlap between the GDM-associated dmCpGs among the different studies is likely to reflect the different GDM diagnostic criteria used, age at analysis, ethnicity, and/or the inclusion of different confounders. Many previous reports did not take account of the confounding influence of maternal prepregnancy BMI and/or found that the signal associated with maternal GDM was largely attenuated by inclusion of maternal BMI. Here, all women had a prepregnancy BMI of over 30 kg/m$^2$, lessening the chance of a confounding effect of BMI, and in a sensitivity analysis, we further adjusted for any potential confounding effect of obesity. More recently, a meta-analysis of 7 pregnancy cohorts [32] (3,677 mother–newborn pairs with 317 GDM cases) identified 2 DMRs associated with GDM within OR2L13 and CYP2E1. There was no overlap with the DMRs identified in this study; however, the GDM diagnostic criteria was different, and we have studied the GDM-associated signal within a high-risk population of women with obesity rather than across the spectrum of BMI, which may account for the greater number of dmCpGs and DMRs associated with GDM diagnosis in this study. Furthermore, we have, for the first time in a GDM study, used the updated Illumina 850K array, which has increased coverage over important regulatory regions and shown that DNA methylation changes associated with GDM and dysglycaemia are enriched among these regions.

There were some limitations to this study. The first was that we compared DNA methylation patterns in cord blood of infants, and the functional significance of such changes are unclear; methylation changes in blood cells could reflect alterations in immune function and/or chronic low grade inflammation which is strongly linked to increased adiposity and insulin resistance. Secondly, the methylation changes observed were modest and the functional significance of these changes is unknown. This is a common finding in epigenetic studies investigating the impact of maternal GDM exposure or other early life environmental factors [23,25,33,52,73]. As DNA methylation is essentially binary with each CpG site being either 0%

or 100% methylated, a small change in CpG methylation observed across a tissue will reflect a change in DNA methylation in a small fraction of cells within the tissue. Depending upon the location of the CpG site, this may affect the level of gene transcription with consequences for cell function. Since DNA methylation changes are heritable, such changes may be perpetuated during subsequent cell division, leading to persistent changes in function/metabolic capacity that could alter susceptibility to disease. However, no RNA was available from the samples to determine whether the methylation changes observed were accompanied by a corresponding variation in gene expression. Nevertheless, the DNA methylation changes reported in this study may prove useful as indices of early life glycaemic exposure. Thirdly, as uncontrolled glucose homeostasis during pregnancy can result in poor pregnancy outcomes, the majority of the women diagnosed with GDM were treated, with the treatment regime predicated on the severity/aetiology of the glycaemia; GDM treatment generally began with dietary advice, followed by metformin with the addition of/replacement with insulin if control was not achieved. Because of the strong collinearity between treatment modality and severity of hyperglycaemia, we cannot exclude treatment as a potential confounder. However, similar numbers of women with GDM in each arm of the trial received treatment through dietary advice, insulin only, metformin only, or a combination of metformin plus insulin, supporting the conclusion that the lifestyle intervention led to attenuation of the infant methylation signal. Finally, generalisability cannot be assumed, and future studies should address the optimum gestational window for intervention and reproducibility in lean pregnant women with GDM. In future, we will examine the associations of these methylation patterns at birth with metabolic function and anthropometric measures of the children at the 3-year follow-up visit, recently completed, and as the children grow to maturity.

## Conclusions

In summary, we found that maternal pregnancy GDM and dysglycaemia led to changes in cord blood DNA methylation, and GDM/dysglycaemia-associated dmCpGs are sensitive to a maternal nutritional and physical activity intervention in the second trimester of pregnancy in women with obesity. The data imply that second trimester lifestyle interventions to improve metabolic health could influence the next generation through persistent influences on the epigenome. In future, we will assess whether the attenuated methylation signature is associated with improved child health outcomes particularly with respect to 25- to 28-week FPG, 1-h, and 2-h PG levels.

## Supporting information

**S1 Data. Supplementary tables.** Table A: List of pyrosequencing primers. Table B: Comparison of samples included in cohort vs samples not included in this analysis. Table C: Total list of dmCpGs associated with maternal GDM. Table D: Total list of dmCpGs associated with maternal GDM after adjustment for maternal BMI. Table E: Total list of dmCpGs associated with 1-h glucose levels. Table F: Total list of dmCpGs associated with 2-h glucose levels. Table G: List of dmCpGs overlapping between the different analyses. Table H: List of dmCpG associated with maternal GDM after adjustment for gestational age at OGTT. Table I: List of dmCpGs asscociated with maternal 1-h glucose levels after adjustment for gestational age at OGTT. Table J: List of dmCpGs associated with GDM and 1-h glucose in the sex-stratified analysis. Table K: Top 20 enriched GO terms from the methylGSA analysis. Table L: Total list of DMRs associated with maternal GDM. Table M: Total list of DMRs associated with 1-h glucose levels. Table N: Total list of DMRs associated with 2-h glucose levels. Table O: List of DMRs overlapping between the different analyses. Table P: List of GDM-associated dmCpGs

overlapping GDM-associated DMRs. Table Q: Total list of dmCpGs associated with 1-h glucose levels after adjustment for intervention. Table R: Total list of dmCpGs associated with GDM after adjustment for intervention. Table S: Total list of dmCpGs associated with GDM in the control arm of the intervention. Table T: Total list of dmCpGs associated with 1-h glucose levels in the control arm of the intervention. Table U: Difference in the effect sizes between the control and intervention arm of the CpG associations with GDM and 1-h glucose levels. Table V: Total list of significant mQTLs with the dmCpGs associated with 1-h glucose levels in the control arm of the intervention.
(XLSX)

**S1 Text. Strenthening the Reporting of Observational Studies in Epidemiology (STROBE) guildelines.**
(DOCX)

**S2 Text. Project analysis plan.**
(DOCX)

**S1 Fig. Enrichment of dmCpGs among histone modifition and regulatory regions.** (A) Enrichment of dmCpG among different chromatin states as determined by the ENCODE Hidden Markov Model in HUVECs. (B) Overlap of dmCpGs with different histone modifications (H). Enrichment was calculated using the Fisher exact test. $^*p < 0.05$, $^{**}p < 0.01$, $^{***}p < 0.001$, $^{****}p < 0.0001$.
(TIF)

**S2 Fig. Fasting, 1-h, and 2-h glucose levels in the participants separated by intervention group.** Comparison of (A) fasting glucose, (B) 1-h glucose, and (C) 2-h glucose levels separated by intervention arm, showing no difference in either measure between the 2 groups.
(TIF)

## Author Contributions

**Conceptualization:** Joanna D. Holbrook, Graham C. Burdge, Lucilla Poston, Keith M. Godfrey, Karen A. Lillycrop.

**Data curation:** Michael S. Kobor.

**Formal analysis:** Elie Antoun, Negusse T. Kitaba, Philip Titcombe, Kathryn V. Dalrymple, Emma S. Garratt, Sheila J. Barton, Robert Murray, Paul T. Seed, David TS Lin, Julia L. MacIsaac, Sara L. White.

**Funding acquisition:** Graham C. Burdge, Lucilla Poston, Karen A. Lillycrop.

**Investigation:** Elie Antoun, Negusse T. Kitaba, Kathryn V. Dalrymple, Robert Murray, Joanna D. Holbrook, Michael S. Kobor, Sara L. White, Karen A. Lillycrop.

**Methodology:** Elie Antoun, Negusse T. Kitaba, Philip Titcombe, Kathryn V. Dalrymple, Emma S. Garratt, Sheila J. Barton, Paul T. Seed, Michael S. Kobor, Graham C. Burdge, Sara L. White, Lucilla Poston, Keith M. Godfrey.

**Project administration:** Joanna D. Holbrook, Lucilla Poston, Keith M. Godfrey, Karen A. Lillycrop.

**Resources:** Kathryn V. Dalrymple, Lucilla Poston, Keith M. Godfrey, Karen A. Lillycrop.

**Supervision:** Joanna D. Holbrook, Lucilla Poston, Keith M. Godfrey, Karen A. Lillycrop.

**Validation:** Emma S. Garratt.

**Visualization:** Elie Antoun.

**Writing – original draft:** Elie Antoun, Negusse T. Kitaba, Lucilla Poston, Keith M. Godfrey, Karen A. Lillycrop.

**Writing – review & editing:** Philip Titcombe, Kathryn V. Dalrymple, Emma S. Garratt, Sheila J. Barton, Robert Murray, Paul T. Seed, Joanna D. Holbrook, Michael S. Kobor, David TS Lin, Julia L. MacIsaac, Graham C. Burdge, Sara L. White.

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
