## [Editor Report · Decision Letter 0]

7 Feb 2020

Dear Dr Lillycrop, 

Thank you for submitting your manuscript entitled "Maternal dysglycemia induces changes in the infant’s epigenome which are ameliorated by a diet and physical activity intervention in pregnancy" for consideration by PLOS Medicine.

Your manuscript has now been evaluated by the PLOS Medicine editorial staff [as well as by an academic editor with relevant expertise] and I am writing to let you know that we would like to send your submission out for external peer review.

Kind regards,

Adya Misra, PhD,

Senior Editor

PLOS Medicine

---

## [Decision Letter · Decision Letter 1]

18 May 2020

Dear Dr. Lillycrop,

Thank you very much for submitting your manuscript "Maternal dysglycemia induces changes in the infant’s epigenome which are ameliorated by a diet and physical activity intervention in pregnancy" (PMEDICINE-D-20-00268R1) for consideration at PLOS Medicine. 

[LINK]

In light of these reviews, I am afraid that we will not be able to accept the manuscript for publication in the journal in its current form, but we would like to consider a revised version that addresses the reviewers' and editors' comments. Obviously we cannot make any decision about publication until we have seen the revised manuscript and your response, and we plan to seek re-review by one or more of the reviewers. 

We expect to receive your revised manuscript by Jun 01 2020 11:59PM. Please email us (plosmedicine@plos.org) if you have any questions or concerns.

We look forward to receiving your revised manuscript. 

Sincerely,

Adya Misra, PhD

Senior Editor 

PLOS Medicine

plosmedicine.org

Please revise your title according to PLOS Medicine's style. Your title must be nondeclarative and not a question. It should begin with main concept if possible. "Effect of" should be used only if causality can be inferred, i.e., for an RCT. Please place the study design ("A randomized controlled trial," "A retrospective study," "A modelling study," etc.) in the subtitle (ie, after a colon).

Abstract 

* Please include the study design, population and setting, number of participants, years during which the study took place, length of follow up, and main outcome measures.

* Please quantify the main results (with 95% CIs and p values). 

* Please include the important dependent variables that are adjusted for in the analyses. 

Abstract Conclusions:

* Please address the study implications without overreaching what can be concluded from the data; the phrase "In this study, we observed ..." may be useful.

* Please interpret the study based on the results presented in the abstract, emphasizing what is new without overstating your conclusions.

Author summary

References should be placed within square brackets please and bibliography should be in Vancouver style 

Methods

Please mention where the original trial took place 

Did your study have a prospective protocol or analysis plan? Please state this (either way) early in the Methods section.

Discussion

Lines 428-432 need to be toned down. Please add “to our knowledge” to avoid assertions of primacy. In addition, since this study can be strictly described as “observational” I would avoid the use of causal language. 

Please ensure that the study is reported according to the STROBE guideline, and include the completed checklist as Supporting Information. When completing the checklist, please use section and paragraph numbers, rather than page numbers. Please add the following statement, or similar, to the Methods: "This study is reported as per the Strengthening the Reporting of Observational Studies in Epidemiology (STROBE) guideline (S1 Checklist)."

Comments from the reviewers:

Reviewer #1: This manuscript capitalises on samples and date generated from women who participated in UPBEAT (a) to address the relationship between maternal glycaemia and cord blood DNA methylation patterns and (b) to examine the ability of a lifestyle intervention (diet and physical activity) to modify these associations. The study has several strengths (including the large numbers, validation of the array data and consideration of heterogeneity) but I have several comments that the authors should consider:

1. The women included in the current study represent about a third of those in the total UPBEAT study. How did the characteristics of the subset differ from the whole cohort and on what basis were they selected (or what determined if a buffy coat from cord blood was collected)?

2. It is mentioned that fetal sex was included in the regression models. However given that there is accumulating evidence for the importance of sex difference in the field, it is surprising that sex specific effects are neither investigated or discussed. 

3. It is a strength that validation of the array data by pyrosequencing was included. What were the % validation rates observed?

4. Was any threshold for methylation differences incorporated in the analysis? Some differences look to be very small - discussion of the potential biological relevance of difference is important.

5. How many of the loci as being identified as being differentially methylated relate to genes that are expressed in cord blood cells? Given the relatively late nature of the intervention, many of its effects on methylation are likely to be tissue specific, therefore discussion of potential functional relevance is important.

6. Line 471- the statement that high 1 hour glucose reflects impaired insulin secretion needs a reference.

7. Line 547 - the term widespread is probably not appropriate - it is actually a small percentage of loci affected

8. Table 1 - glucose and insulin measurements are probably not accurate to two decimal places.

Reviewer #2: Thanks for the opportunity to review your manuscript. My role is as a statistical reviewer, so I have concentrated my review around the data and analysis (and the presentation of these). I should declare that I do not have bioinformatics expertise and I recommend that someone with expertise in this area review the manuscript. I have included general comments and queries to begin with, followed by questions that are more specific.

This study is a secondary analysis of data from a randomised control trial comparing usual care with a diet and physical activity intervention in pregnant women with a BMI >30. The main study found that the intervention was successful in improving diet and physical activity, but there were no differences detected in the primary outcomes of GD development and LGA infants. In this secondary study the authors have examined whether DNA methylation levels from infants were associated with maternal GD status (and other continuous glycaemic measures) 

This is an interesting study and there are some improvements I think could be made to the current manuscript. As PLOS Medicine has a general audience, it would be helpful for an explanation of the key outcomes (i.e. the measurements of methylation) to be included so that the importance of this work is obvious to someone without extensive experience in genomics.

Was a protocol or analysis plan created for this secondary study? If this were available, it would be helpful in reviewing.

How was missing data accounted for in the analysis? The rates of missingness are not extremely high, but there are key measurements (fasting insulin, HBA1c etc.) with a reasonable level of missing data and my impression from a look at the tables is that this may depend on GD diagnosis. Was a complete-case analysis used or a listwise deletion strategy? 

L193. 'Hierarchical clustering' is vague, what exact algorithm and metrics were used for this analysis?

L214. What form did the covariates take in the regression model (continuous, polynomial, or as dummy variables)?

How were assumptions (linearity, heteroscedacity, distribution of residuals) checked?

L218. More details are needed about the sensitivity analysis - how exactly were these conducted? Are these sub-group analyses or a type of moderation/mediation model?

L218. FDR is a much more utilised concept for controlling for multiplicity in genomic analysis as opposed to clinical/epidemiology where we are much more used to familywise methods. I do not think the application of the B-H method here is inappropriate but it does achieve power at the cost of higher Type I rates. Is FDR generally accepted as appropriately for looking at how patient level measurements are associated with levels of methylation?

L253/Figure S1. Fisher's exact test can be used where all the margins of the tables are fixed, which I don't think is the case with this data. There are alternatives e.g. Barnard's test for when 1 or more margins aren't fixed.

L355. Is there are way to present point estimates/effect size measures from these linear models in addition to just the p-values? I

Table 1. The summary information here is useful but I don't think that the p-values are useful, given that a null hypothesis of no difference isn't realistic given the groups are stratified according to GDM diagnosis. I would just remove the p-values. 

Reviewer #3: The current study investigated the impacts of maternal GDM and continuous glucose levels on cord blood DNA methylation and additionally examined how lifestyle interventions that improve maternal glycemic control influence these relationships. Major strengths of the study include the design, sample size, and the acquisition of genetic data, which allowed the authors to evaluate how genetic variation influences their findings. However, there are also some important limitations. In particular, I do not believe that the statistical analyses performed were appropriate for testing the authors' hypothesis that the lifestyle treatments modify the impacts of GDM/glucose on cord blood methylation. Several key references were also missing from the Introduction and Discussion. I also have concerns about the methods used to conduct the pathway analyses. Some additional minor concerns are also described in more detail below.

Introduction:

Line 73: There is a typo in the first sentence of the Introduction

Lines 95-96: Although most studies on GDM/glucose and methylation have been small, the PACE consortium recently published a meta-analysis on GDM and cord blood methylation, which merits some brief discussion in the Introduction (PMID: 31601636).

Lines 96-98: This statement is not true: "all have focussed on GDM versus no GDM, rather than the continuous relationship between maternal glucose levels and DNA methylation". Some previous studies have evaluated maternal glucose continuously in relation to offspring methylation (e.g., see PMID: 29752424 and 22396200). There is also a typo in this sentence.

Methods:

Lines 188-189: "CpGs known to cross-hybridise to other locations in the genome (n=14,759), coinciding with SNPs (n=77,261)". Please cite the appropriate supporting reference here.

Lines 214-218: Please provide some detail about how covariates were selected

Were results similar after additional adjustment for gestational age at OGTT? This needs to be addressed.

Network analyses: Do these analyses account for the fact that some genes are overrepresented on the EPIC array? There are software and methods available that account for this that may be more appropriate 

Results:

Were any of the CpGs still statistically significant after applying the more conservative Bonferroni correction? This is worth reporting.

Did the DMRs identified using DMRcate overlap with the differentially methylated CpGs identified using the traditional EWAS single CpG approach? Or were there notable differences? It would be interesting to see more details on this.

Lines 389-391: "The increase in number of dmCpGs associated particularly with 1h-PG levels after adjustment for intervention suggested that the intervention might attenuate the 1h-PG associated methylation signature in the infant". This rationale isn't clear to me. Wouldn't you expect to see fewer differentially methylated CpGs after adjusting for treatment? Also, if effect modification is of interest, the main comparison should be the results stratified by treatment arm (i.e., Table 2). However, rather than conducting an EWAS for these stratified analyses, it would make more sense to identify CpGs that are significantly differentially methylated by glucose level in the control group and then compare the magnitudes of the effect estimates for these same loci in the intervention arms. 

Discussion:

Lines 421-422 "There is increasing evidence that maternal dysglycemia has adverse effects on the health of the offspring, predisposing the infant to developing obesity and metabolic disease in later life." Please provide supporting references for this statement.

Lines 461-462 "higher maternal 1h-PG levels concentrations reflect impaired insulin secretion". Please provide supporting references for this statement.

Additional limitations: 

Please be explicit about the lack of gene expression data

Please address generalizability as a limitation

Please address the lack of follow-up data on methylation in childhood as a limitation (or a potential future direction)

Lines 519-521: "More recently, a meta-analysis of seven pregnancy cohorts (3,677 mother-newborn pairs with 317 GDM cases) identified 2 DMRs associated with GDM within OR2L13 and CYP2E1". The citation for this study is missing.

[LINK]

---

## [Decision Letter · Decision Letter 2]

17 Aug 2020

Dear Dr. Lillycrop,

Thank you very much for re-submitting your manuscript "Maternal dysglycaemia, changes in the infant’s epigenome and amelioration by a diet and physical activity intervention in pregnancy:  secondary analysis of a randomised control trial" (PMEDICINE-D-20-00268R2) for review by PLOS Medicine.

I have discussed the paper with my colleagues and the academic editor and it was also seen again by reviewers. I am pleased to say that provided the remaining editorial and production issues are dealt with we are planning to accept the paper for publication in the journal.

[LINK]

We look forward to receiving the revised manuscript by Aug 21 2020 11:59PM. 

Sincerely,

Adya Misra, PhD

Senior Editor 

PLOS Medicine

plosmedicine.org

Requests from Editors:

["... amelioration by a diet ... intervention" in the title is bit too declarative. Please revise to "Maternal dysglycaemia, changes in the infant’s epigenome modified with a diet and physical activity intervention in pregnancy: secondary analysis of a randomised control trial. 

Overall: can you please change all instances of “obese” to “with obesity”. 

Please adapt the text of the abstract (we suggest around line 77) to note that further research will be needed to investigate possible medical implications of the findings. 

Please provide brief participant demographics in the methods and findings section

Please provide exact FRD values in all instances, unless below 0.001

Line 639- do you mean to say lower glycemic excursions? Please revise as needed

Please add a space between text and reference brackets throughout 

“652 DNA Methylation can be influenced by the genotype of the individual, with sequence variation 653 at specific loci resulting in different patterns of DNA methylation[72]. These sites are called 654 methylation quantitative trait loci (meQTLs) and contribute to inter individual differences in 655 DNA methylation and differential response to environmental factors” perhaps this information should be provided in the Introduction?

Line 721- you may wish to revise to “In future, we will examine the associations …” and the same in line 731 

The main manuscript doesn’t need information about funding or competing interests as these are provided in the article meta-data

At line 108 and several other points in the paper, you state that "methylation changes ... were reduced by ... the intervention". Although such language is appropriate in reporting the main outcomes of appropriately powered RCTs, here we note that the observations were not primary outcomes of the study, and dropouts may also influence the findings. Therefore, we ask you to adopt more cautious language at all relevant instances in the paper, e.g., "methylation changes ... appeared to be reduced ...".

We note that no significant effects were reported on the UPBEAT trial's two primary outcomes - please amend the text around line 159 to indicate this. 

Please make that "strengthening" at line 192.

Please correct the typo(s) at line 308. 

At line 309, please note any changes made to the analysis plan after the study had begun. 

Please adapt the text around line 726 to "led to changes" and so on.

Comments from Reviewers:

Reviewer #1: No additional comments - my original comments have all been addressed.

Reviewer #2: Thanks for the revised manuscript and detailed responses to my original queries. The additional text in the discussion was helpful for me to understand the methylation measurements. The additions to the methods section have clarified all of my original queries and I think these changes have made the analysis easier to understand as well. 

I agree the using an alternative to Fisher's may not make much difference here, and given that a conservative bias especially problematic with these measurement (i.e. these are not safety outcomes from a RCT) so this does not need any further changes. 

Table S2 is a helpful addition along with the text in the discussion. I still wasn't clear what would have made a patient more likely to have a buffy coat sample taken. i.e. is this linked to provision of certain clinical procedures, or were buffy-coat samples only taken at particular sites or at certain phases of the study?

Reviewer #3: The authors have done an excellent job addressing my previous concerns. I believe the manuscript is now suitable for publication.

[LINK]

---

## [Editor Report · Decision Letter 3]

6 Oct 2020

Dear Professor Lillycrop, 

On behalf of my colleagues and the academic editor, Dr. Ronald C. W. Ma, I am delighted to inform you that your manuscript entitled "Maternal dysglycaemia, changes in the infant’s epigenome modified with a diet and physical activity intervention in pregnancy:  secondary analysis of a randomised control trial." (PMEDICINE-D-20-00268R3) has been accepted for publication in PLOS Medicine. 

PRODUCTION PROCESS

Before publication you will see the copyedited word document (within 5 busines days) and a PDF proof shortly after that. The copyeditor will be in touch shortly before sending you the copyedited Word document. We will make some revisions at copyediting stage to conform to our general style, and for clarification. When you receive this version you should check and revise it very carefully, including figures, tables, references, and supporting information, because corrections at the next stage (proofs) will be strictly limited to (1) errors in author names or affiliations, (2) errors of scientific fact that would cause misunderstandings to readers, and (3) printer's (introduced) errors. Please return the copyedited file within 2 business days in order to ensure timely delivery of the PDF proof. 

If you are likely to be away when either this document or the proof is sent, please ensure we have contact information of a second person, as we will need you to respond quickly at each point. Given the disruptions resulting from the ongoing COVID-19 pandemic, there may be delays in the production process. We apologise in advance for any inconvenience caused and will do our best to minimize impact as far as possible.

PRESS

PROFILE INFORMATION

Thank you again for submitting the manuscript to PLOS Medicine. We look forward to publishing it. 

Best wishes, 

Adya Misra, PhD

Senior Editor 

PLOS Medicine

plosmedicine.org